# Epigenome-wide meta-analysis identifies DNA methylation biomarkers associated with diabetic kidney disease

Laura J. Smyth [1,20], Emma H. Dahlström [2,3,4,20], Anna Syreeni [2,3,4], Katie Kerr[1], Jill Kilner [1], Ross Doyle[5], Eoin Brennan [5], Viji Nair[6], Damian Fermin[7], Robert G. Nelson[8], Helen C. Looker[8], Christopher Wooster [1], Darrell Andrews[5], Kerry Anderson[1], Gareth J. McKay[1], Joanne B. Cole [9,10], Rany M. Salem [11], Peter J. Conlon[12], Matthias Kretzler [13], Joel N. Hirschhorn[9,14,15], Denise Sadlier[16], Catherine Godson[5], Jose C. Florez [9,10,17], GENIE consortium*, Carol Forsblom [2,3,4], Alexander P. Maxwell [1,18], Per-Henrik Groop [2,3,4,19], Niina Sandholm [2,3,4] ✉ & Amy Jayne McKnight [1] ✉

Type 1 diabetes affects over nine million individuals globally, with approximately 40% developing diabetic kidney disease. Emerging evidence suggests that epigenetic alterations, such as DNA methylation, are involved in diabetic kidney disease. Here we assess differences in blood-derived genome-wide DNA methylation associated with diabetic kidney disease in 1304 carefully characterised individuals with type 1 diabetes and known renal status from two cohorts in the United Kingdom-Republic of Ireland and Finland. In the meta-analysis, we identify 32 differentially methylated CpGs in diabetic kidney disease in type 1 diabetes, 18 of which are located within genes differentially expressed in kidneys or correlated with pathological traits in diabetic kidney disease. We show that methylation at 21 of the 32 CpGs predict the development of kidney failure, extending the knowledge and potentially identifying individuals at greater risk for diabetic kidney disease in type 1 diabetes.

Type 1 diabetes (T1D) is a chronic disease characterised by hyperglycaemia due to insulin deficiency resulting from autoimmune beta-cell destruction[1]. Approximately 40% of individuals with diabetes develop diabetic kidney disease (DKD), a microvascular diabetic complication, which remains one of the most common causes of chronic kidney disease (CKD) worldwide[2,3]. Individuals with diabetes have an increased risk of developing kidney failure, necessitating renal replacement therapy in the form of dialysis or transplantation and raising overall morbidity and mortality[4]. Accumulating evidence indicates that epigenetic factors play a role in T1D[5] and the development and progression of DKD among individuals with T1D[6].

Epigenetic modifications provide a link between an individual's genetics and the environment to which they are exposed. These alterations can regulate gene expression without altering the DNA base sequence and have been associated with several complex diseases, including T1D and type 2 diabetes (T2D) and kidney diseases[7,8]. Over time, powerful epigenome-wide association studies (EWAS), utilising blood-derived DNA, have revealed associations between methylation levels and DKD. Using the previously developed Illumina methylation array, Infinium HumanMethylation450 Beadchip, including 485,000 methylation sites, Qui et al.[9] identified 77 differentially methylated CpG sites associated with a decline in estimated glomerular filtration rate (eGFR) in 181 Pima Indians with diabetes and chronic kidney disease. The more recently developed array—the MethylationEPIC BeadChip—provides more comprehensive coverage, including the methylation status of additional 413,745 sites. Using this array, Sheng

A full list of affiliations appears at the end of the paper. *A list of authors and their affiliations appears at the end of the paper.
✉e-mail: niina.sandholm@helsinki.fi; a.j.mcknight@qub.ac.uk

et al.[10] identified seven differentially methylated CpGs associated with albuminuria or eGFR in 500 individuals with kidney disease and diabetes of unspecified type from the Chronic Renal Insufficiency Cohort (CRIC). In 2021, we[11] identified 36 differentially methylated CpGs associated with kidney failure in 360 individuals with T1D using the same array.

Here, we conducted an EWAS on DKD, including 1,304 carefully phenotyped individuals with T1D from the United Kingdom and Republic of Ireland (UK-ROI) and the Finnish Diabetic Nephropathy Study (FinnDiane) using the higher-density Illumina Infinium MethylationEPIC array. We aimed to assess differences in DNA methylation in whole blood between individuals with DKD attributed to T1D and those with long duration T1D and no evidence of kidney disease on repeated testing.

## Results
### Study cohorts
This study included 1304 participants with T1D (651 cases and 653 controls with and without DKD) from the United Kingdom−Republic of Ireland (UK-ROI, $n = 504$) and the Finnish Diabetic Nephropathy (FinnDiane, $n = 800$) study. Cases had DKD defined as persistent macroalbuminuria, i.e., albumin excretion rate (AER) > 300 mg/ml in urine, and controls had an AER within the normal range despite a long duration of T1D (≥15 years). Participant characteristics by cohort and DKD are included in Table 1, with summary data, including data missingness, available in Supplementary Table 1. Due to case-control matching, no differences were observed between cases and controls for age, sex, diabetes duration and smoking (Table 1).

### Quality control of generated data and pre-processing
We used the Illumina Infinium MethylationEPIC BeadChip array to examine 866,895 CpG sites for all samples. Houseman estimates[12] were calculated for the proportional white cell counts (WCCs) for each sample (Table 1). Concordance of methylation β-values for seven duplicate samples was evaluated, with a mean $r^2$ of 0.99 (Supplementary Fig. 1). Following the quality control (QC) and pre-processing steps, including sex checks and the assessment of cross-reactive probes conducted by RnBeads, one to seven individuals and 90,391 to 102,252 probes were removed prior to the analysis depending on the model and cohort (Supplementary Fig. 2, Supplementary Table 2).

### Meta-analysis of differentially methylated CpGs from RnBeads analysis
We evaluated differential methylation levels between cases and controls of DKD using Rnbeads, in EWASs conducted separately in both cohorts. The false discovery rate (FDR) adjusted p-values from both cohorts were combined in a sample-size weighted meta-analysis using METAL[13]. Altogether methylation at 32 CpGs exceeded the threshold of $p \leq 9.9 \times 10^{-8}$ (Table 2), required for epigenome-wide significance, in at least one of the three analysis models. The minimal model ($n = 1302$ after QC), which adjusted for age, sex and WCCs, identified methylation at 31 CpGs associated with DKD ($p \leq 9.9 \times 10^{-8}$, Supplementary Data 1, Fig. 1a). The majority of these differentially methylated CpGs were located within genes (Fig. 2a) or in the open sea region in relation to the nearest CpG Island (Fig. 2b). Most of the identified CpGs were hypomethylated, i.e., had a lower methylation level in DKD cases than controls ($n = 26$). Of the DKD-hypomethylated CpGs, the largest difference was observed at cg03546163, located in the 5′UTR of the *FKBP5* gene (Methylation β-value difference=0.062, $p = 3.6 \times 10^{-13}$). Only six CpGs were hypermethylated, i.e., had a higher methylation level in DKD (Fig. 3a), with the largest difference observed for cg17944885 located between *ZNF788P* and *ZNF625-ZNF20* (Methylation β-value difference=0.069, $p = 2.0 \times 10^{-44}$). In the association model with additional adjustment for current smoking status ($n = 1175$), methylation

levels at 18 CpGs differed between cases and controls of DKD ($p \leq 9.9 \times 10^{-8}$, Figs. 1b, 2b and Supplementary Data 1). These were also significant in the minimally adjusted model, except cg19693031−a previously known differentially methylated CpG for HbA$_{1c}$ in type 1 diabetes in the 3′UTR of the *TXNIP* gene[14]. The number of epigenome-wide associated differentially methylated CpGs was reduced to seven in the maximally adjusted model ($n = 957$), which included glycated haemoglobin (HbA$_{1c}$), HDL cholesterol, triglycerides, body mass index (BMI) and duration of diabetes ($p \leq 9.9 \times 10^{-8}$, Figs. 1c, 2c and Supplementary Data 1). All seven epigenome-wide significant differentially methylated CpGs in the maximal model were associated with DKD ($p \leq 9.9 \times 10^{-8}$) in the other models. The QQ plots showed no presence of inflation (Fig. 1), but instead we observed some deflation of the p-values, particularly in the maximal model (Fig. 1c), which was likely due to the FDR adjustment of the p-values and the multiple covariate adjustments in that model.

### DKD-associated differentially methylated CpGs and the development of kidney failure
To evaluate whether the methylation levels at the 32 DKD-associated CpGs predicted the risk of kidney failure in T1D, we conducted prospective analyses including 397 individuals with DKD at the time of DNA collection and follow-up data available from the FinnDiane study. In the survival models adjusted for age, sex, and WCCs, methylation at 21 of the 32 DKD-associated CpGs predicted the development of kidney failure (nominal $p < 0.05$; 10 with a Bonferroni-adjusted $p < 1.56 \times 10^{-3}$; Table 2, Supplementary Data 2). Methylation at cg17944885, located between genes *ZNF788P* and *ZNF625-ZNF20*, represented the strongest signal associated with the risk of kidney failure in individuals with macroalbuminuria (HR [95% CI] = 2.31[1.95, 2.76], $p = 1.40 \times 10^{-21}$). The estimated effects of all significant differentially methylated CpGs were in the same directions in the prospective analysis as in the cross-sectional EWAS analysis. When we added the baseline eGFR to the survival model, methylation of the two CpGs on chromosome 19 remained significantly associated with the risk of kidney failure (Bonferroni-adjusted $p < 0.03$; Supplementary Data 2). Moreover, when adding CpG methylation to a survival model with eight clinical risk factors for kidney failure (age, sex, age at diabetes onset, systolic blood pressure, HbA$_{1c}$, triglycerides, smoking, retinal photocoagulation) the model prediction improved significantly for 13/32 CpGs ($p < 0.05$ for model concordance improvement).

### DKD-associated differentially methylated CpG associations in external EWAS on DKD
All 32 epigenome-wide significant differentially methylated CpGs were searched in available, non-overlapping summary EWAS data on DKD, which included an EWAS on albuminuria, eGFR, eGFR slope, and HbA$_{1c}$ performed in 473 individuals with any diabetes from the CRIC study cohort[10] (Table 2, Supplementary Table 3). Significant differential methylations were observed for eleven CpGs ($p < 0.05$), of which five were significant after Bonferroni correction (32 CpGs × four phenotypes). These included four differentially methylated CpGs associated with eGFR; a highly significant cg21961721 within *SLC27A3* ($p < 2.2 \times 10^{-308}$), two intergenic CpGs between *ZNF788P* and *ZNF625-ZNF20* (cg17944885; $p = 3.7 \times 10^{-13}$ and cg25544931; $p = 2.1 \times 10^{-5}$), and cg05165263 within the *IRF2* gene ($p = 6.6 \times 10^{-5}$). Methylation at none of the CpGs was associated with eGFR slope or albuminuria after Bonferroni correction. For HbA$_{1c}$, only cg19693031 within *TXNIP* remained significant after Bonferroni correction ($p_{Bonferroni} = 8.0 \times 10^{-12}$).

### Potential overlaps of differentially methylated CpGs with transcription factors
To assess the potential functional consequences of the identified differentially methylated CpGs, we searched the eFORGE-TF database for

**Table 1 | Clinical characteristics for the UK-ROI and FinnDiane participants by DKD cases and controls**

| | UK-ROI | | | | FinnDiane | |
| | Cases ($n = 252$) | Controls ($n = 252$) | p | Cases ($n = 399$) | Controls ($n = 401$) | p |
|---|---|---|---|---|---|---|
| Sex, Male | 132 (52.4) | 132 (52.4) | 1.00 | 151 (37.7) | 149 (37.3) | 0.99 |
| Age, years | 42.1 (9.4) | 41.2 (9.6) | 0.30 | 43.2 (10.8) | 43.6 (11.0) | 0.62 |
| Diabetes onset age, years | 15.3 (7.3) | 15.1 (7.3) | 0.92 | 13.0 (8.1) | 14.4 (8.6) | 0.02 |
| Diabetes duration, years | 27.5 (7.3) | 26.2 (7.4) | 0.17 | 30.2 (9.3) | 29.2 (9.0) | 0.13 |
| Smoking status, current | 39 (25.8) | 50 (22.1) | 0.17 | 102 (25.4) | 101 (25.3) | 1.00 |
| HbA$_{1c}$, % | 9.3 (2.3) | 8.2 (1.4) | $5.5 \times 10^{-11}$ | 8.9 (1.6) | 8.1 (1.2) | $5.8 \times 10^{-13}$ |
| HbA$_{1c}$, mmol/mol | 78 (25) | 66 (15) | $5.5 \times 10^{-11}$ | 74 (18) | 65 (13) | $5.8 \times 10^{-13}$ |
| BMI, kg/m$^2$ | 25.7 (5.1) | 28.9 (8.0) | 0.004 | 26.7 (4.5) | 25.4 (3.3) | $5.0 \times 10^{-6}$ |
| Estimated GFR, mL/min/1.73m$^2$ | 19.9 (19.8) | 90.0 (20.3) | $<1.0 \times 10^{-20}$ | 54.0 (29.9) | 91.8 (16.9) | $<1.0 \times 10^{-20}$ |
| HDL cholesterol, mmol/mol | 1.6 (0.6) | 1.6 (0.4) | 0.25 | 1.27 (0.4) | 1.39 (0.4) | $5.2 \times 10^{-5}$ |
| Triglycerides, mmol/mol | 2.2 (1.9) | 1.3 (1.2) | 0.006 | 1.79 (1.2) | 1.05 (0.5) | $<1.0 \times 10^{-20}$ |
| CD8 T cells, proportion | 0.040 (0.042) | 0.041 (0.044) | 0.71 | 0.056 (0.047) | 0.048 (0.041) | 0.007 |
| CD4 T cells, proportion | 0.130 (0.071) | 0.139 (0.057) | 0.13 | 0.111 (0.048) | 0.101 (0.050) | 0.003 |
| Natural killer cells, proportion | 0.028 (0.039) | 0.044 (0.048) | $1.5 \times 10^{-11}$ | 0.045 (0.046) | 0.029 (0.043) | $7.2 \times 10^{-7}$ |
| B cells, proportion | 0.026 (0.025) | 0.037 (0.025) | $2.3 \times 10^{-6}$ | 0.040 (0.027) | 0.030 (0.026) | $4.8 \times 10^{-7}$ |
| Monocytes, proportion | 0.085 (0.040) | 0.079 (0.033) | 0.04 | 0.075 (0.028) | 0.079 (0.030) | 0.03 |
| Granulocytes, proportion | 0.666 (0.124) | 0.625 (0.100) | $2.0 \times 10^{-4}$ | 0.631 (0.091) | 0.674 (0.087) | $2.8 \times 10^{-7}$ |

Continuous variables are reported as mean (standard deviation), and categorical variables are reported as number (%). The p-values for continuous variables are from a two-sided t-tests and for categorical variables from chi-squared tests (one-sided). Proportions refer to the proportion in white blood cells.

*AER* Albumin excretion rate, *BMI* body mass index, *HbA$_{1c}$* haemoglobin A1c, *HDL* high-density lipoprotein, *n* number, *ROI* Republic of Ireland, *SD* standard deviation, *UK* United Kingdom, *WCCs* white cell counts.

transcription factor motifs overlapping with the epigenome-wide significant CpGs using the seven kidney datasets[15]. Seven CpGs overlapped with predicted transcription factor binding sites (Table 2). These included two CpGs with higher methylation in DKD that overlapped with predicted binding sites for the transcription factor Pax-3 (cg05165263 in *IRF2*) and the Vitamin D3 receptor (cg10072464 between *GRHL1-KLF11*; Supplementary Table 4). In addition, five of the hypomethylated CpGs in DKD cases overlapped with predicted transcription factor binding sites for the following factors (Supplementary Table 4): the homeodomain leucine zipper-containing factor (cg22815707 in *ANKRD12*), the peroxisome proliferator-activated receptor gamma (cg17058475 in *CPT1A*), the sterol regulatory element-binding protein 1 (cg19996939), the transcription factor AP-2 alpha (cg05710777 within *LINC10800*) and the p53 and transcriptional activator Myb (both overlapped with cg01895164).

### Correlations between DKD-associated differentially methylated CpGs and gene expression

Next, we examined the association between the 32 DKD-associated differentially methylated CpGs and mRNA expression of nearby genes from available summary data of *cis*-expression quantitative trait methylation sites (*cis*-eQTMs) in monocytes ($n = 1202$)[16], whole-blood of 2102 participants from Dutch biobank[17] and 4,170 participants from the Framingham Heart Study[18] (only *cis*-eQTMs with $p < 1.0 \times 10^{-8}$) and in human kidney samples ($n = 414$)[19]. Methylation at three of DKD-associated CpGs were associated with the gene expression in cis in monocytes, five in whole blood, and ten in kidneys (Supplementary Data 3). Of these, methylations at six CpGs were associated with the expression of the gene they were located in (Supplementary Data 3, Table 3). The differentially methylated CpG with the lowest *p*-value in the EWAS on DKD, cg17944885, was associated with a lower expression of several nearby zinc finger genes in both monocytes and whole blood; the lowest *p*-value was observed for the *ZNF844* gene ($p = 3.6 \times 10^{-26}$). Two CpGs were associated with the expression of the same nearest gene in both whole blood and kidneys: cg17058475 within the *CPT1A* and cg19693031 within the *TXNIP* gene. Higher methylation at these two sites, both hypomethylated in DKD vs controls, was

associated with lower expression of genes *CPT1A* and *TXNIP*, respectively, in both whole blood and kidneys.

### Assessment of gene expression profiles in kidney tissue in additional datasets

As DNA methylation is known to regulate gene expression, we next investigated if the 21 genes containing DKD-associated differentially methylated CpGs were differentially expressed in kidneys in additional external datasets from the Nephroseq v4 database, the North Dublin Renal Biobank and Pima Indians (Table 3).

First, we searched these genes in the Nephroseq database for differential gene expression. Ten genes were differentially expressed in kidney biopsies from individuals with DKD (Table 3, Supplementary Table 5). Of the genes with hypermethylated CpGs in DKD vs controls, only one gene was differentially expressed in DKD; the *SLC27A3* gene had a 1.6-fold higher expression in kidney glomeruli of individuals with DKD compared to healthy living donors ($p = 6.61 \times 10^{-5}$)[20]. Of the genes with DKD-associated CpGs with lower methylation in DKD, *FKPB5* gene expression was reduced in all six kidney datasets, including gene expression data from kidney glomeruli and tubuli. *ANKRD12, NME7* and *REV1* showed a 1.6 to 2.5-fold decreased expression in kidney glomeruli from individuals with DKD compared to healthy living donors ($p < 0.01$) in the Woroniecka Diabetes Dataset[21]. In the same dataset, *GRK5* expression was 4.3-fold decreased in kidney glomeruli ($p = 9.49 \times 10^{-7}$) and 1.5-fold increased in kidney tubules ($p = 0.026$). Furthermore, *INPP4B, MBNL1* and *PTBP3* showed a higher gene expression in kidney tubuli with a fold change ranging from 1.5 (for *INPP4B*, $p = 0.012$) to 2.6 (for injection *PTBP3*, $p = 1.2 \times 10^{-5}$). Four alterations in gene expression were recorded for *TXNIP*; the most significant was in the Woroniecka Diabetes Glomeruli Dataset ($p = 1.84 \times 10^{-4}$, Fold change: $-1.5$)[21].

Next, gene expression levels of genes containing DKD-associated differentially methylated CpGs were investigated in the North Dublin Renal Biobank. These data are derived by RNASeq analysis of renal biopsy material from CKD patients. In total, 11 genes showed a difference in gene expression after adjustment for participant age and sex ($p < 0.05$, Table 3, Supplementary Table 6). Among these, higher *TXNIP* expression was associated with progression to kidney failure

**Table 2 | Differentially methylated CpGs from the three meta-analysis models and associations from prospective analyses, CRIC study and eFORGE-TF database**

| CpG Site | Gene | Minimal model ($p$) | Minimal + Smoking model ($p$) | Maximal model ($p$) | Progression to kidney failure ($p$) | DKD CRIC cohort ($p$) | eFORGE-TF ($p$) |
|---|---|---|---|---|---|---|---|
| cg19693031 | *TXNIP* | $6.75 \times 10^{-7}$ | $8.80 \times 10^{-8}$ | $6.17 \times 10^{-7}$ | | $6.22 \times 10^{-14}$ (HbA$_{1c}$) | |
| cg05284887 | *GJA5* | $3.35 \times 10^{-9}$ | $2.60 \times 10^{-7}$ | $1.42 \times 10^{-5}$ | $3.93 \times 10^{-5}$ | | |
| cg21961721 | *SLC27A3* | $2.34 \times 10^{-14}$ | $5.57 \times 10^{-11}$ | $2.75 \times 10^{-10}$ | $1.72 \times 10^{-4}$ | $<2.2 \times 10^{-308}$ (eGFR) | |
| cg08150816 | *NME7* | $8.23 \times 10^{-15}$ | $3.71 \times 10^{-13}$ | $4.62 \times 10^{-10}$ | $7.16 \times 10^{-5}$ | | |
| cg01895164 | *PAFAH2, EXTL1* | $9.06 \times 10^{-12}$ | $1.79 \times 10^{-8}$ | $7.79 \times 10^{-6}$ | | | $1.08 \times 10^{-6}$ |
| cg23527387 | *REV1* | $4.12 \times 10^{-11}$ | $2.80 \times 10^{-8}$ | $2.26 \times 10^{-3}$ | $5.19 \times 10^{-3}$ | | |
| cg02841972 | *GRHL1, KLF11* | $1.35 \times 10^{-8}$ | $4.65 \times 10^{-8}$ | $1.53 \times 10^{-6}$ | | | |
| cg05710777 | *LINC01800* | $2.95 \times 10^{-18}$ | $4.05 \times 10^{-15}$ | $1.98 \times 10^{-11}$ | $9.78 \times 10^{-4}$ | | $3.58 \times 10^{-6}$ |
| cg12864625 | *MBNL1* | $1.16 \times 10^{-10}$ | $1.62 \times 10^{-7}$ | $9.98 \times 10^{-6}$ | | | |
| cg18376497 | *INPP4B* | $1.78 \times 10^{-10}$ | $3.15 \times 10^{-9}$ | $1.56 \times 10^{-6}$ | $3.81 \times 10^{-4}$ | 0.005 (eGFR) | |
| cg05165263 | *IRF2* | $9.87 \times 10^{-8}$ | $3.64 \times 10^{-6}$ | $8.18 \times 10^{-5}$ | $1.04 \times 10^{-3}$ | $6.62 \times 10^{-5}$ (eGFR) | $5.42 \times 10^{-6}$ |
| cg12378834 | *C5orf66* | $4.04 \times 10^{-10}$ | $3.06 \times 10^{-7}$ | $2.43 \times 10^{-4}$ | | | |
| cg11414254 | *ZNF346, UIMC1* | $4.02 \times 10^{-8}$ | $2.34 \times 10^{-6}$ | $3.63 \times 10^{-4}$ | | | |
| cg19996939 | *HBS1L, MYB* | $3.40 \times 10^{-8}$ | $2.16 \times 10^{-8}$ | $2.84 \times 10^{-6}$ | | | $2.91 \times 10^{-6}$ |
| cg02917536 | *TAB2* | $9.28 \times 10^{-8}$ | $9.77 \times 10^{-7}$ | $2.73 \times 10^{-4}$ | | 0.018 (eGFR) | |
| cg03546163 | *FKBP5* | $3.63 \times 10^{-13}$ | $8.90 \times 10^{-10}$ | $1.77 \times 10^{-4}$ | | 0.003 (HbA$_{1c}$) | |
| cg00008629 | *PTBP3* | $5.41 \times 10^{-15}$ | $1.17 \times 10^{-12}$ | $2.90 \times 10^{-9}$ | | | |
| cg15167811 | *PTBP3* | $3.03 \times 10^{-11}$ | $2.93 \times 10^{-9}$ | $6.84 \times 10^{-7}$ | | | |
| cg13125822 | *GRK5* | $4.93 \times 10^{-10}$ | $1.64 \times 10^{-8}$ | $1.89 \times 10^{-6}$ | | | |
| cg03026982 | *NAV2* | $1.43 \times 10^{-9}$ | $3.20 \times 10^{-7}$ | $5.30 \times 10^{-5}$ | | | |
| cg10473623 | *NAV2* | $1.63 \times 10^{-8}$ | $1.85 \times 10^{-6}$ | $2.31 \times 10^{-4}$ | | | |
| cg05325763 | *CPT1A* | $2.08 \times 10^{-11}$ | $4.83 \times 10^{-9}$ | $1.93 \times 10^{-7}$ | | | |
| cg17058475 | *CPT1A* | $5.90 \times 10^{-8}$ | $1.07 \times 10^{-6}$ | $2.68 \times 10^{-5}$ | | 0.009 (ALB) | $9.52 \times 10^{-6}$ |
| cg08230697 | *STAB2* | $1.82 \times 10^{-8}$ | $1.23 \times 10^{-6}$ | $5.19 \times 10^{-5}$ | | | |
| cg10072464 | *ADPRHL1* | $9.09 \times 10^{-10}$ | $3.02 \times 10^{-7}$ | $4.96 \times 10^{-7}$ | $1.04 \times 10^{-3}$ | 0.002 (eGFR) | $2.44 \times 10^{-7}$ |
| cg24382141 | *PSKH1* | $3.44 \times 10^{-8}$ | $1.01 \times 10^{-5}$ | $1.46 \times 10^{-3}$ | | 0.002 (eGFR) | |
| cg22815707 | *ANKRD12* | $1.16 \times 10^{-9}$ | $1.70 \times 10^{-8}$ | $4.82 \times 10^{-6}$ | | | $6.31 \times 10^{-6}$ |
| cg25544931 | *ZNF763, ZNF433-AS1* | $2.58 \times 10^{-27}$ | $1.85 \times 10^{-21}$ | $7.90 \times 10^{-19}$ | $4.01 \times 10^{-6}$ | $2.09 \times 10^{-5}$ (eGFR) | |
| cg17944885 | *ZNF788P, ZNF625-ZNF20* | $1.97 \times 10^{-44}$ | $1.38 \times 10^{-36}$ | $8.91 \times 10^{-27}$ | $1.40 \times 10^{-21}$ | $3.72 \times 10^{-13}$ (eGFR) | |
| cg06587767 | *PIP5K1C* | $2.08 \times 10^{-9}$ | $9.47 \times 10^{-7}$ | $2.98 \times 10^{-4}$ | | | |
| cg02711608 | *SLC1A5* | $8.89 \times 10^{-12}$ | $2.62 \times 10^{-9}$ | $4.77 \times 10^{-8}$ | | | |
| cg12230203 | *PTGIS, B4GALT5* | $7.79 \times 10^{-8}$ | $1.25 \times 10^{-6}$ | $5.20 \times 10^{-6}$ | $6.63 \times 10^{-6}$ | | |

The CpG methylations included met the EWAS significance threshold (corrected $p \leq 9.9 \times 10^{-8}$) in one of the three meta-analysis adjustment models. *P*-values in the meta-analysis were derived from hierarchal linear models including the covariates age, sex and six WCCs (minimal model), or age, sex, six WCCs and current smoking status (Minimal + smoking model) or age, sex, six WCCs, current smoking status, HbA$_{1c}$, HDL cholesterol, triglycerides, duration of diabetes and body mass index (Maximal model). For intergenic CpGs, the two closest downstream and upstream genes are given. Empty cells did not have supporting evidence ($p < 1 \times 10^{-5}$) for transcription factors using eFORGE-TF. Empty cells also indicate that no evidence achieved at least $p \leq 0.05$ from the CRIC database lookup. The phenotype with lowest *p*-value shown for the CRIC cohort, which include individuals with any type of diabetes.
Abbreviations
*CRIC* Chronic Renal Insufficiency Cohort Study, *eGFR* estimated glomerular filtration rate, *ALB* Albuminuria, *HbA$_{1c}$* Haemoglobin A$_{1c}$, *T2D* type 2 diabetes, *TF* transcription factor.

($p = 0.004$) and with fibrosis ($p = 0.002$, $n = 83$). Furthermore, *NAV2* expression was associated C-reactive protein ($p = 0.001$, $n = 39$).

Additional lookups were conducted for the same gene list within kidney gene expression data from an American Indian population of Pima Indians with T2D using correlation coefficients for both glomerular and tubular kidney tissue (Supplementary Table 7, Table 3). The expression of five genes in kidney glomerular tissue, including *INPP4B, MBNL1, PTBP3, REV1* and *SLC27A3*, were correlated with various kidney morphology parameters ($p < 0.01$), and the higher expression of three genes in kidney tubular tissue (*GRK5, IRF2* and *STAB2*) correlated with lower eGFR and eGFR slope ($p < 0.01$).

## Functional analyses of gene ontology, pathways, and protein networks

Enrichment analyses for pathways and genomic locations were conducted using all differentially methylated CpGs that reached $p < 1 \times 10^{-5}$

in the meta-analysis and were directionally consistent in both cohorts. These included a total of 119 CpGs. In the genomic region enrichment analyses (Fig. 2c), DKD-associated CpGs were enriched in open sea regions ($p = 5.5 \times 10^{-3}$) and underrepresented in CpG Islands ($p = 2.8 \times 10^{-7}$) and within 200 bp from the transcription start site ($p = 1.4 \times 10^{-3}$). For the gene ontology (GO) and Kyoto Encyclopedia of genes and genome (KEGG) pathway analyses, the 119 CpGs were annotated to 77 genes. The three top enriched GO pathways (Fig. 4a) for the annotated genes were *alkali metal ion binding* ($p = 5.6 \times 10^{-5}$, GO:0031420), *transferase activity, transferring phosphorus−containing groups* ($p = 3.8 \times 10^{-4}$, GO:0016772) and *kinase activity* ($p = 4.4 \times 10^{-4}$, GO:0016301). The three top enriched KEGG pathways (Fig. 4b) included *primary bile acid biosynthesis* ($p = 9.8 \times 10^{-4}$, path:hsa00120), *insulin resistance* ($p = 9.8 \times 10^{-3}$, path:hsa04931) and *intestinal immune network for IgA production* ($p = 0.01$, path:hsa04672). However, none of the pathways was significant after controlling for a 5% FDR.

**Table 3 | Supporting evidence from eQTM data (whole-blood, monocytes, and kidney) and gene expression data from kidney biopsies for the genes containing DKD-associated CpGs (gene expressions with $p < 0.05$ reported)**

| Gene (CpG; methylation DKD cases-controls) | CpG methylation effect on gene expression (eQTM) | Gene expression (FC) in DKD (Nephroseq) | Whole-kidney gene expression in CKD (NDRBB) | Glomerular gene expression and kidney morphometry correlations (r) (Pima T2D) |
|---|---|---|---|---|
| ANKRD12 (cg22815707;−2.3%) | | Glom:−1.6 (W) | | Intact podocyte foot processes, % (r = 0.3), endothelial fenestrations, % (r = 0.3) |
| C5orf66 (cg12378834;−6.1%) | | | CRP (log$_2$R = 0.36) | |
| FKBP5 (cg03546163; −0.9%) | Kidney: Lower CpG methylation increases FKBP5 | Glom:−3.2 to −1.6 (Ju), TubInt:−3.3 to −1.5 (W, S, ERCB) | | |
| GJA5 (cg05284887;−4.0%) | | | eGFR (log$_2$R = −0.80), fibrosis (log$_2$R = −0.68) | GBM width (r = −0.3), Intact podocyte foot processes (r = 0.2), Peripheral GBM surface vol. (r = 0.3) |
| GRK5 (cg13125822;−0.8%) | | Glom:−4.3 (W), TubInt: 1.5 (W) | | Peripheral GBM surface vol. (r = −0.3) |
| INPP4B (cg18376497; −2.3%) | Whole blood: Lower CpG methylation increases INPP4B | TubInt:1.5 (W) | eGFR (log$_2$R = −0.48), fibrosis (log$_2$R = −0.71) | Podocyte vol. per glomerulus (r = −0.3), Podocyte cell per glomerulus, n (r = −0.2) |
| IRF2 (cg05165263; +4.2%) | | | Fibrosis (log$_2$R = 0.30) | |
| MBNL1 (cg12864625; −0.2%) | | TubInt: 1.6 (W) | Fibrosis (log$_2$R = 0.53), kidney failure (log$_2$R = 0.42) | Podocyte cell per glomerulus, n (r = −0.3) |
| NAV2 (cg03026982;−3.5%) | | | CRP (log$_2$R = −0.31) | GBM width (r = 0.3), Peripheral GBM surface vol. (r = −0.3) |
| NME7 (cg08150816; −1.8%) | | Glom: −2.5 (W) | | |
| PIP5K1C (cg06587767;−3.4%) | | | | Endothelial fenestrations % (r = −0.2), Mesangial fractional vol. (r = 0.3) |
| PSKH1 (cg24382141; −1.6%) | | | eGFR (r = 0.41) | |
| PTBP3 (cg00008629; −3.2%) | Kidney: Lower CpG methylation increases PTBP3 | | Fibrosis (log$_2$R = 0.49), eGFR (r = −0.58) | Podocyte foot process width (r = −0.3) |
| REV1 (cg23527387; −3.4%) | | Glom:1.6 (W) | | |
| SLC1A5 (cg02711608; −1.4%) | Monocytes, whole blood: Lower CpG methylation increases SLC1A5 | | eGFR (r = 0.68) | Endothelial fenestrations % (r = −0.3) |
| SLC27A3 (cg21961721; 1.3%) | | Glom:1.6 (Ju) | eGFR (r = −0.47) | Podocyte cell per glomerulus, n (r = −0.3) |
| TAB2 (cg02917536; −2.8%) | | | Fibrosis (log$_2$R = 0.3) | Endothelial fenestrations % (r = 0.3) |
| TXNIP (cg19693031; −3.0%) | Kidney, monocytes, whole blood: Lower CpG methylation increases TXNIP | Glom:−1.5 (W), TubInt:1.6 to 3.1(W, S) | Fibrosis (log$_2$R = 0.81), Kidney failure (log$_2$R = 0.97), eGFR (r = −0.68) | Intact podocyte foot processes % (r = −0.3), endothelial fenestrations % (r = 0.2) |

The methylation in DKD was calculated as the methylation percentage β value difference between cases and controls (β value in cases minus β value in controls), average of UK-ROI and FinnDiane methylation levels. Nephroseq results (Fold changes) were based on seven kidney datasets. Woroniecka (W) datasets included results from 22 glomerular (9 with DKD) and 22 tubular (10 with DKD) kidney samples collected from healthy, living transplant donors and diagnostic kidney biopsies. The Schmid (S) datasets (n = 2) included the tubulointerstitium of 24 kidney biopsies (13 with DKD). The Ju datasets (n = 2) included results from kidney biopsies of 12 individuals with DKD, which were compared to healthy living donors or other diseases. The ERCB (the European Renal cDNA Bank-Kroener-Fresenius biopsy bank) included 19 kidney samples from the tubulointerstitium (10 with DKD). The Northern Dublin Renal Bank (NDRBB) included whole-kidney biopsies from 44 individuals with multiple CKD etiologies. Reported log$_2$ ratios calculated in the R package 'limma' and log$_2$-ratios have an FDR-adjusted p-value <0.05. Correlations are Spearman's correlations.

DKD diabetic kidney disease, CKD chronic kidney disease, GFR glomerular filtration rate, cytosine-phosphate-guanine, NDRBB North Dublin Renal Biobank, T2D type 2 diabetes, vol volume, GBM Glomerular basement membrane. Glom Glomeruli, TubInt Tubulointerstitium.

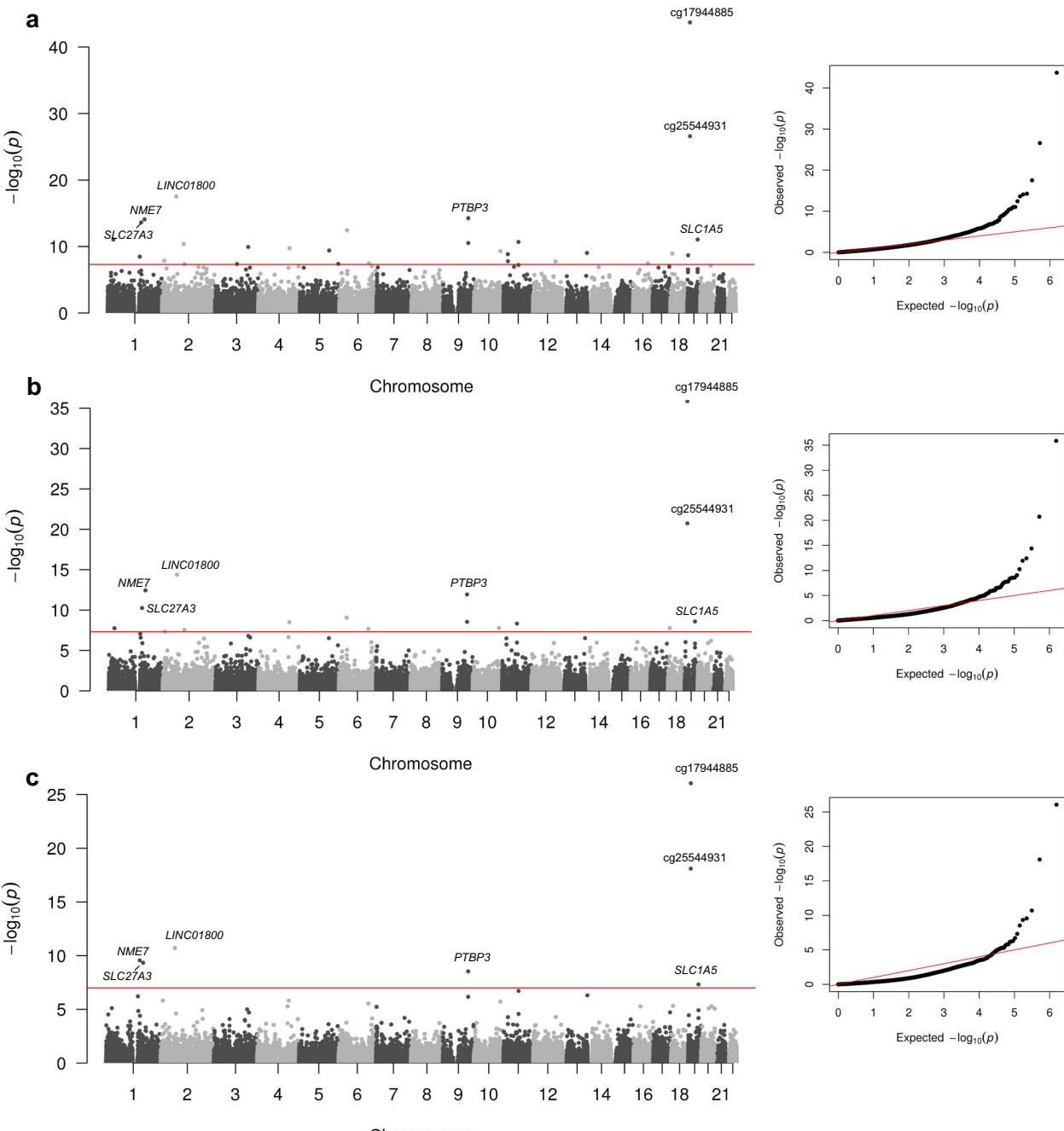

**Fig. 1 | Manhattan and qq-plots of the associations from the epigenome-wide association meta-analysis on DKD in type 1 diabetes using three different analysis models (a, b and c).** The models were adjusted for (**a**) age, sex and six WCCs (Minimal model), (**b**) Minimal model and current smoking status, and (**c**) minimal model and current smoking status, HbA$_{1c}$, high-density lipoprotein cholesterol, triglycerides, duration of diabetes and body mass index. The x-axis shows the chromosomal locations, and the y-axis shows −log$_{10}$(p-values). P-values in the EWAS (FinnDiane and UK-ROI) were derived from hierarchal linear models and FDR-adjusted, after which they were meta-analysed using METAL. The red line indicates the epigenome-wide significance threshold ($p \leq 9.9 \times 10^{-8}$). Gene symbols (or CpG identifier if intergenic) are displayed for differentially methylated CpGs that reached epigenome-wide significance across all models.

## Overlap with CpGs detected in previous EWAS studies

To compare overlap with previous epigenome-wide association studies on DKD, we looked up CpG methylations previously associated with DKD-related phenotypes (Kidney failure, eGFR, albuminuria) in epigenome-wide association studies performed in the blood[9–11] or kidney tubuli[22] in our meta-analysis. We observed some overlap with results from the CRIC study[10]; 13 (14%) of the eGFR-associated CpGs in that study were also associated with DKD in our study (Supplementary Data 4). Also, 43 differentially methylated CpGs that we previously identified for kidney failure in type 1 diabetes[11] were associated with DKD in this study. Methylation levels of three CpGs in kidney tubuli (out of 65) associated with interstitial fibrosis[22], were also differentially methylated in our meta-analysis on DKD. No overlap was observed with the findings from Qiu et al.[9] for EWAS on kidney failure in Pima Indians. To further evaluate the overlap with other traits, we performed trait enrichment analysis for DKD-associated CpGs ($p < 1 \times 10^{-5}$) against the reported associations in the EWAS atlas. The top 20 most significantly associated traits are shown in Fig. 4c. Trait enrichment analyses revealed significant overlap with several established risk factors for DKD, such as ageing, blood pressure, eGFR and smoking.

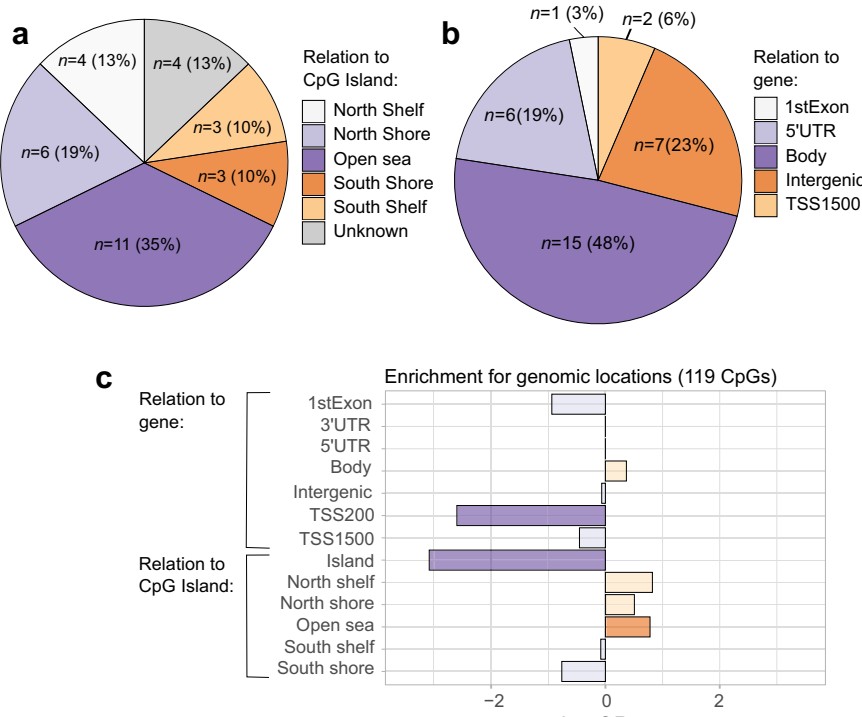

**Fig. 2 | The location of the 31 DKD-associated CpGs in the minimal model ($p \leq 9.9 \times 10^{-8}$) in relation to the CpG island (a) or the gene (b) and enriched/depleted genomic locations for CpGs with $p < 1.0 \times 10^{-5}$ in the meta-analysis minimal model.** Locations in panel **a** and **b** were retrieved from Illuminas Infinium MethylationEPIC v1.0 B4 Manifest file. The genomic enrichment analyses in panel **c**, including DKD-associated CpGs with a $p < 10^{-5}$, was performed within the web-based EWAS atlas platform. Significant results ($p < 0.05$) are denoted by darker shades. Enriched regions (orange colour) were Open Sea ($p = 5.5 \times 10^{-3}$), and depleted regions (purple colours) were TSS200 ($p = 1.4 \times 10^{-3}$) and Island ($p = 2.8 \times 10^{-7}$). The EWAS toolkit uses the Weighted Fisher's Exact test (two-sided) to compute $p$-values. *DKD* Diabetic kidney disease, *TSS* transcription starting site, *UTR* Untranslated region.

## Mendelian Randomisation

To evaluate the causal effect of the methylation at CpGs associated with DKD, we performed two-sample Mendelian randomisation using known methylation quantitative trait loci (mQTL) in whole-blood[23]. In total, seven blood mQTLs were available for the 32 DKD-associated CpGs, including cg23527387 (*REV1*), cg19693031 (*TXNIP*), cg18376497 (*INPP4B*), cg17944885 (between *ZNF788P* and *ZNF625-ZNF20*), cg00008629 (*PTBP3*), cg03546163 (*FKBP5*) and cg10072464 (*ADPRHL1*). Of these, genetically determined higher methylation levels at cg23527387 decreased the risk of DKD (*REV1*; causal OR = 0.74 (0.58-0.94)), $p = 0.015$, Fig. 5, Supplementary Table 8). For CpGs with several instruments available (mQTLs for CpGs in or near *ADPRHL1*, *INPP4B* and *ZNF625-ZNF20*), we found no evidence of pleiotropy using the heterogeneity or Egger intercept test (Supplementary Table 8).

## Meta-analysis of CpG islands, genes, promoters, and genomic regions

Differential methylation levels were aggregated and compared on a regional level for CpG islands, genes, promoters, and genomic tiling regions (window size = 5 kb), identifying one gene and five tiling regions associated with DKD ($p \leq 9.9 \times 10^{-8}$). In the gene-based analysis, the *RP4-800F24.1*, a long noncoding RNA gene located within the *NME7* gene on chromosome 1 (Supplementary Table 9), was significant in both the minimally adjusted model ($p = 1.09 \times 10^{-9}$) and on adjustment by smoking status ($p = 1.34 \times 10^{-8}$). The five tiling regions associated with DKD were located within *NME7*, *C5orf66*, *PTBP3*, and *STAB2* genes and close to *ZNF20*, of which the tiling region within *PTBP3* was significant across each of the three models (Supplementary Data 5). Supplementary Table 9 and Supplementary Data 5 additionally include results that exceeded a $p \leq 10^{-5}$ threshold.

## Discussion

This research was conducted to assess alterations in blood-derived DNA methylation patterns associated with DKD in individuals with a long duration of T1D in the largest meta-analysis of DKD in T1D performed to date. DKD in T1D was associated with altered blood DNA methylation at 32 CpGs sites across the genome. Most of these CpGs were hypomethylated in DKD compared to controls with a long duration of T1D and no evidence of kidney disease. Of these, 25 CpGs were within genes and seven were epigenome-wide significant in all adjusted models; cg00008629 (*PTBP3*), cg02711608 (*SLC1A5*), cg05710777 (*LINC01800*), cg08150816 (*NME7*), cg21961721 (*SLC27A3*) and the intergenic cg17944885 and cg25544931 (Supplementary Fig. 3). Furthermore, we show that methylation levels at 21 of these sites predicted the development of kidney failure in T1D ($p < 0.05$) and found evidence for causal effects for the methylation of cg23527387 within the *REV1* gene.

The most significantly associated differentially methylated CpG—with the largest effect on methylation in DKD—was observed at cg17944885 situated in a cluster of zinc finger genes on chromosome 19, between *ZNF788P* and *ZNF20-ZNF625*. A second non-gene centric differentially methylated CpG, cg25544931, also located in the same gene cluster nearby *ZNF763* and *ZNF433-AS1*, similarly reached epigenome-wide significance in all adjusted models. Both sites were hypermethylated in the blood of individuals with DKD and appeared co-methylated (Spearman $r = 0.56$ [95%CI: 0.51–0.60] in FinnDiane, $p = 2.3 \times 10^{-66}$). Additionally, in our prospective sub-analyses, an increase in methylation at cg17944885 strongly predicted the progression of DKD, from macroalbuminuria to kidney failure (HR = 2.3). Hypermethylation at cg17944885 has previously been associated with reduced eGFR in males with the human immunodeficiency virus[24], as well as reduced eGFR and increased risk of CKD and albuminuria in an EWAS meta-analysis in the general population[25–27]. Hypermethylation

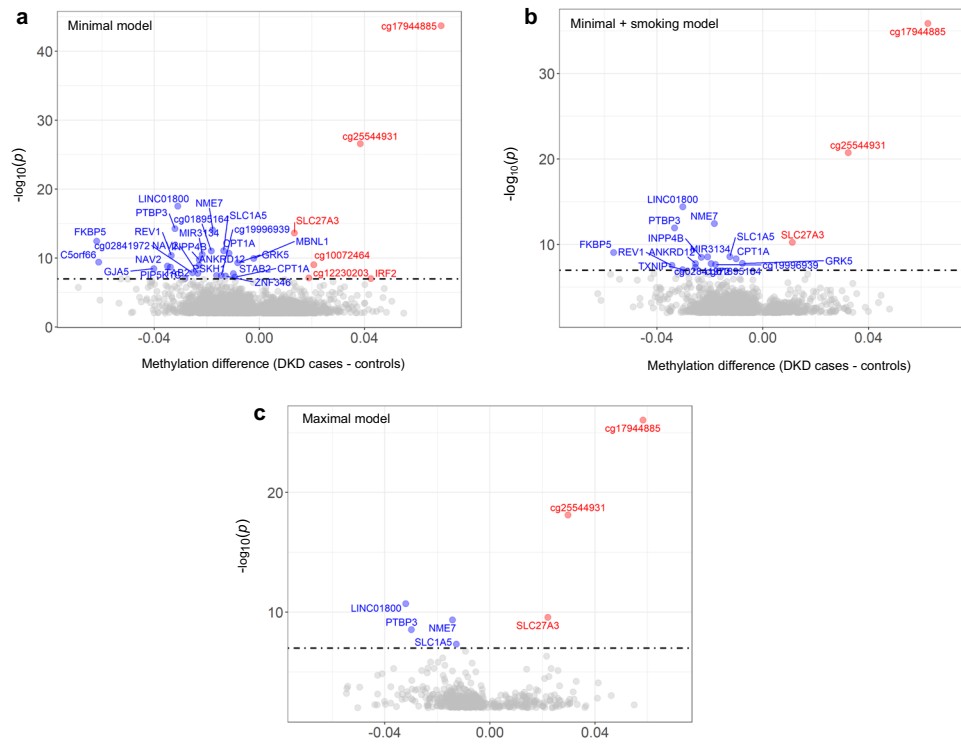

**Fig. 3 | Volcano plot of methylation differences for CpGs (DKD cases vs controls without DKD) in the meta-analyses of the minimal model (a), minimal model + smoking (b), and maximal model (c).** The red colour denotes differentially methylated CpGs with $p < 9.9 \times 10^{-8}$ and hypermethylated in DKD cases vs controls and the blue colour denotes differentially methylated CpGs with $p < 9.9 \times 10^{-8}$ and hypomethylated in DKD cases vs controls. The dotted line indicates the epigenome-wide significance threshold ($p \leq 9.9 \times 10^{-8}$). The x-axis shows the mean methylation difference (=mean methylation β-values in DKD cases−mean methylation β-values in DKD controls) and the y-axis shows −log10(p-values). P-values were from the meta-analysis, which combined FDR-adjusted p-values from the individual EWASs with p-values derived from hierarchical linear models. Minimal model included covariates age, sex and six white cell counts. Minimal + smoking model included covariates age, sex, six WCCs and current smoking status. Maximal model included covariates age, sex, six WCCs, current smoking status, HbA$_{1c}$, HDL cholesterol, triglycerides, duration of diabetes and body mass index.

of cg17944885 has also been reported previously in DKD; in individuals with established DKD ($n = 38$) compared to individuals with early stages of DKD ($n = 83$, CKD stages 1-3a)[28] and reduced eGFR in 500 individuals with DKD[10]. Although the closest gene to cg17944885 is *ZNF20-ZNF265*, it was recently reported that methylation levels at this site correlate with gene expression of a more distant zinc finger protein-coding gene: *ZNF439*[26]. In addition, higher methylation of cg17944885 is associated with the repressed transcription of several other nearby zinc finger genes in whole blood (Supplementary Data 3).

The largest effect on methylation observed among the sites hypomethylated in DKD in this study was observed for cg03546163 within the *FKBP5* gene. Methylation at this CpG reached epigenome-wide significance in all EWAS models except the maximally adjusted EWAS ($p = 1.77 \times 10^{-4}$). Nevertheless, *FKBP5* gene expression was significantly reduced in DKD in all but one of the seven kidney datasets in the Nephroseq database, further supporting the role of this gene in DKD. The *FKBP5* gene encodes the protein FKBP51, an important negative regulator of Akt phosphorylation[29] and NF-κB activation[30]. It is also involved in glucocorticoid receptor signalling[31]. Furthermore, genome-wide analyses of human blood have found associations between higher *FKBP5* mRNA and a pro-inflammatory profile[32]. Hypomethylation at cpg03546163 has previously been associated with DKD ($p = 2.4 \times 10^{-9}$)[33] and kidney failure[11] in T1D. Furthermore, differential methylation of *FKBP5* is associated with multiple other diseases, including T2D and cardiometabolic risk[34].

Five other CpGs reached epigenome-wide significance in all three models. One of these was the cg05710777 located in a long noncoding gene, *LINC01800*, with no previous links to DKD. However, the CpG

site overlapped with a binding site for the transcription factor AP-2 alpha, which has been implicated in the pathogenesis of DKD[35]. Two other differentially methylated CpGs associated with DKD in all three models were cg08150816 in the *NME7* gene and cg00008629 in the *PTBP3* gene. Methylation at cg08150816 in the *NME7* gene, which encodes for a nucleoside-diphosphate kinase required for efficient nucleation of the microtubules in the cell[36], has not previously been linked to any trait. Hypomethylation at cg00008629 in *PTBP3* has been associated with several traits, such as hypertension in 17,010 participants from the CHARGE consortium[37], aging in a study of 42 Hainan centenarians[38], smoking[39] and recently also to kidney disease in the general population[26]. *PTBP3* encodes an RNA binding protein that likely functions as part of the hematopoietic system[40].

The final two CpGs associated with DKD across all models were both located within solute carrier genes; cg21961721 in *SLC27A3* and cg02711608 in *SLC1A5*. *SLC27A3*, which encodes a protein involved in lipid metabolism, was highly significant also with eGFR in the CRIC cohort ($p < 1.0 \times 10^{-303}$) and higher gene expression in the kidney correlated with both DKD and a lower eGFR. *SLC1A5* encodes an amino acid transporter, and methylation at cg02711608 within this gene has been associated with T2D[41], glucose metabolism[42], triglycerides[43], alcohol consumption[44], and recently also with serum urate levels[45]. Blood pressure is also thought to influence the methylation status of cg02711608[46]. Moreover, genetic variants within the *SLC1A5* gene, including rs10412340 and rs402072, have been associated with a genetic predisposition to T1D[47].

Furthermore, the Mendelian randomisation analyses suggested that higher methylation levels at cg23527387 were also causally related

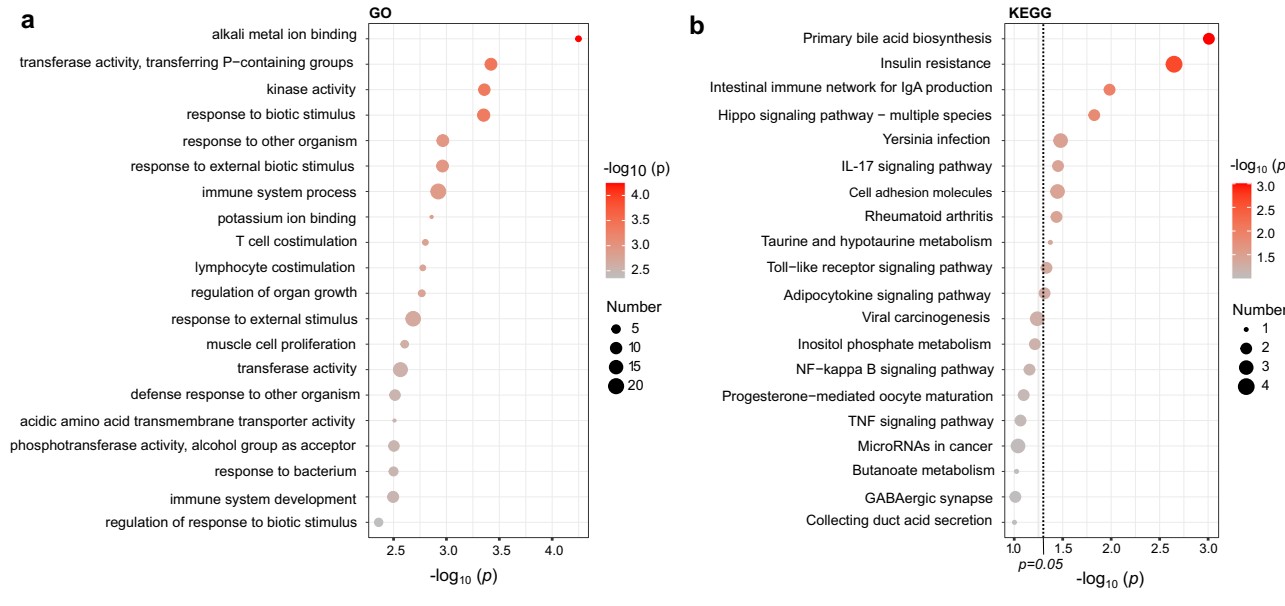

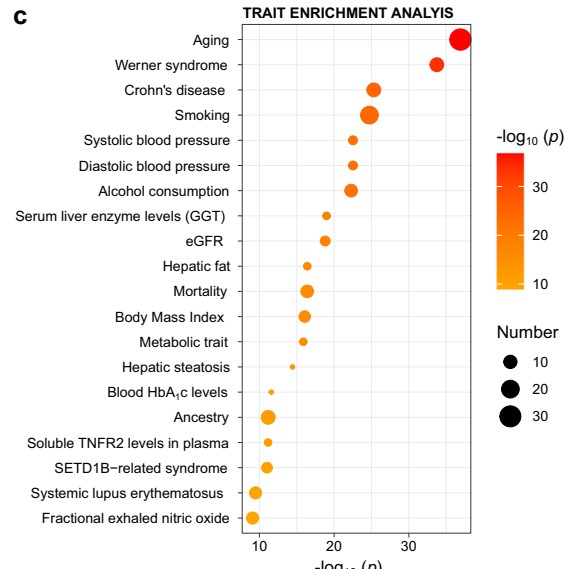

**Fig. 4 | Gene ontology (GO) and Kyoto Encyclopaedia of Genes and Genomes (KEGG) enrichment analyses for genes annotated to CpGs with a $p < 1.0 \times 10^{-5}$ and enriched trait associations in the EWAS Atlas for the individual CpGs.** The size of the points reflects the number of differentially methylated genes associated with each term (panel **a** and **b**) or the number of CpGs identified for this phenotype in the EWAS Atlas (panel **c**). The colour gradient reflects the association strength. The *y*-axis shows $-\log_{10}(p\text{-values})$. The dotted line corresponds to $p = 0.05$ and *p*-values were derived from a hypergeometric test implemented in the missMethyl R package. TNFR2 tumour necrosis factor receptor 2, GGT Gamma–glutamyl transferase (GGT), eGFR estimated glomerular filtration rate, TEA Trait enrichment analysis, P phosphorus.

to a lower risk of DKD, in line with the direction seen in the EWAS. Hypomethylation of cg23527387 has also been observed in a previous methylation analysis on DKD in 226 individuals with T1D using data generated from the Infinium 450 K methylation array ($\beta = 0.86$, $p_{\text{FDRadj}} = 6.0 \times 10^{-7}$)[33]. Cg23527387 is situated on chromosome 2 within the *REV1* gene (Supplementary Fig. 4), which encodes the DNA repair protein REV1, reported to recruit DNA polymerases to damaged DNA (Supplementary Data 6). Although this gene has not previously been linked to DKD in T1D, a genetic variant (rs7583877) in the neighbouring gene, *AFF3*, has been associated with kidney failure in T1D (OR = 1.29, $p = 1.2 \times 10^{-8}$, Supplementary Fig. 4)[48].

## Strengths and limitations
Overall, this investigation had many strengths. This EWAS was conducted to identify variation in methylation status between individuals

with T1D-DKD and individuals with T1D and no evidence of kidney disease across three countries, recruited with the same inclusion and exclusion criteria. With 1,304 participants, this is the largest EWAS conducted to date on T1D-DKD. Furthermore, we had follow-up data available for 397 cases with DKD, which allowed us to evaluate which of the identified differentially methylated CpGs were associated with progression to kidney failure. Two independent cohorts, UK-ROI and FinnDiane, were included with closely matched phenotypic characteristics using the highest-density methylation array available (Infinium MethylationEPIC). Methylation status was assessed using blood-derived DNA from both cohorts that were adjusted for proportional WCCs and cell heterogeneity. Three models considered different potential confounding variables, including a minimal model adjusted for age, sex and six WCCs; the minimal model plus current smoking status, and the fully adjusted model that further considered HbA_{1c},

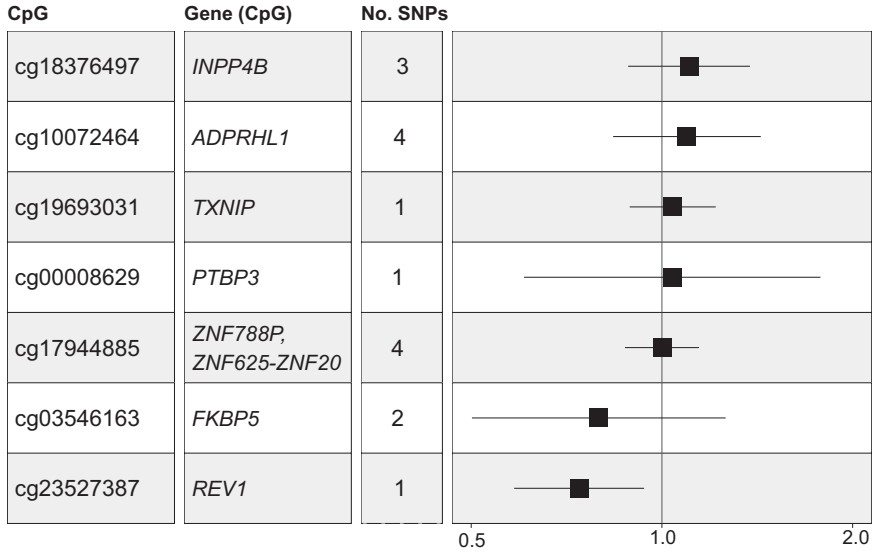

**Fig. 5 | Forest plots detailing the Two-sample MR estimates (ORs) from the association between DNA methylation at seven CpGs and DKD.** The ORs were calculated using the inverse-variance weighted method if several mQTL were available for the CpG site or the Wald ratio if only one mQTL was available. The x-axis shows the OR and its 95% confidence interval for DKD per SD change in DNA methylation for each CpG. OR Odds ratio, DKD Diabetic kidney disease.

HDL cholesterol, triglycerides, duration of diabetes and BMI—recognised risk factors for T1D-DKD.Furthermore, having meta-analysed the results from the two cohorts, only the differentially methylated CpGs, which met the threshold of $p \leq 9.9\times10^{-8}$ and were directionally consistent in both cohorts were considered significant. This threshold was previously reported to reduce the rate of false-positives in studies which use the Infinium MethylationEPIC array[49]. Epigenome-wide significant differentially methylated CpGs and genes have additionally been further examined using platforms including Nephroseq and in complementary cohorts such as the North Dublin Renal Biobank.

Using DNA methylation levels measured from the blood instead of kidneys to make inferences about changes in the kidneys is a potential limitation. The lifespan of the cells in kidneys vs blood is broadly different, which might impact the observed methylation changes. For example, podocytes—the highly specialised cells of the kidney glomerulus that line the outer surface of the glomerular capillaries—are terminally differentiated, whereas the peripheral blood cells are dividing cells with an average lifespan of 120 days. We cannot completely elude whether the changes occurring in peripheral blood reflect the changes that are acquired throughout the lifespan of the podocyte. However, kidney biopsies are invasive procedures that are performed in individuals with T1D only if there is a clinical suspicion that a non-diabetic kidney disease is likely; therefore, such samples are rarely available. In addition, methylation levels across tissues in an individual are highly correlated ($r = 0.86$)[50] and we have previously shown that DNA methylation changes observed in blood were reflected in kidney biopsies[33]. However, owing to the very specialised and varied phenotype of the cells in the kidneys, there is a possibility that some changes in the methylation levels accompanying diabetic kidney disease are not evident in our study. Future studies will have to characterise these changes and their relevance in diabetic kidney disease. Nevertheless, studying the DNA methylation in accessible tissue such as blood enables the identification of biomarkers that could be used in a routine clinical setting or population-based cohort, where biopsies are not feasible.

There is an imperative need for larger cohorts with diabetes and epigenome-wide methylation data. Although some larger EWAS studies on kidney disease in the general population have been performed recently[26], finding comparably sized cohorts with EWAS data and

diabetes, particularly type 1 diabetes, remains challenging. In the lack of such studies, we were limited to finding additional support for our DKD-associated methylation differences in an external cohort with unspecified diabetes and a different phenotype for DKD (eGFR and albuminuria as quantitative measures), complemented by gene expression data. A larger scale, multi-omics analysis, which includes genetic variation, epigenetic alterations and gene expression analyses on the same samples, would further refine markers of interest and improve understanding of the biological mechanisms of DKD. Additional future studies should consider ethnic diversity beyond the predominantly white population considered here, in addition to kidney biopsy matched to blood-derived comparisons, where possible. Furthermore, the methylation levels at each CpG site may reflect the cause or the consequence of kidney disease. To assess potential causality, we performed Mendelian randomisation, but mQTL data were available for only seven of 32 DKD-associated CpG sites, leaving out 78% of the DKD-associated CpG methylations. Whether methylation at these CpG sites causally relates to DKD has yet to be explored.

In summary, this EWAS meta-analysis performed on two well-characterised cohorts identified several epigenetic signatures associated with DKD in T1D and highlighted several genes not previously associated with kidney disease in T1D. These findings provide future studies with potential links between the genes and the milieu in DKD. The prospective analyses additionally showed that these epigenetic signatures have the potential to identify individuals at increased risk of DKD and progressing to kidney failure. As they are potentially reversible, they offer the possibility of therapeutic intervention if they prove to be causal for DKD, as methylation of CpG within the *REV1* gene did. More comprehensive meQTL data is needed to allow more systematic MR analyses for causality as well as functional studies assessing the biological underpinnings of these loci.

## Methods
### UK-ROI collection
Participants were recruited as part of the All Ireland-Warren 3-Genetics of Kidneys in Diabetes (GoKinD) United Kingdom (UK) Collection[48,51,52] or the Belfast Renal Transplant Collection. All participants were from the UK or Republic of Ireland (ROI) and provided written informed consent for research. There was no financial

compensation. DNA was frozen at −80 °C in multiple aliquots following extraction from whole blood using the salting out method[53] and normalised using PicoGreen quantitation[54] using a CytoFluor® Series 4000 (Applied Biosystems, Thermo Fisher Scientific, CA, USA). Individuals with both T1D and DKD were defined as cases ($n = 252$). These individuals had ≥10 years duration of T1D alongside a diagnosis of DKD defined as persistent macroalbuminuria (≥500 mg/24 hr), eGFR <60 mL/min/1.73m² calculated using the Chronic Kidney Disease Epidemiology Collaboration (CKD-EPI) creatinine equation, retinopathy, and hypertension (systolic/diastolic blood pressure ≥135/85 mmHg). Individuals in the control group had ≥15 years duration of T1D and no evidence of kidney disease on repeat testing i.e., they all had normal urinary albumin excretion and eGFR >60 mL/min/1.73m² ($n = 252$). Control individuals had normal blood pressure and were not taking any anti-hypertensive medication. All participants ($n = 504$) were of European white ancestry. Each case individual was matched to a control individual for sex, ethnicity, diabetes duration to within 5 years, and age to within 10 years. The ethical approval reference number for the UK-ROI GoKinD samples is MREC 175/23 (RO321) and for the Belfast Renal Transplant samples is ORECNI 08/NIR03/79, 12/NI/0003, 12/NI/0178.

### FinnDiane collection
All Finnish participants were recruited as part of the FinnDiane Study, a multicentre study from 93 study centres across Finland. At the study visit, each participant provided written informed consent and underwent a thorough clinical examination. There was no financial compensation. Blood samples were drawn for DNA extraction and measurements of blood lipids, creatinine, and HbA$_{1c}$ as described previously[55]. Participants completed standardised questionnaires with the attending physician to document medical history, including information on comorbidities and smoking. Medical files and laboratory databases were reviewed and all available data on AER collected. All case individuals ($n = 401$) had macroalbuminuria defined as AER > 300 mg/24 h (24 h urine) or > 200 µg/min (overnight urine) in at least two of three consecutive urine collections. In addition, participants were required to have an macroalbuminuric (overnight) or high microalbuminuric (AER ≥ 150 mg/24 h) sample at the study date when DNA was collected. Case individuals were matched with controls ($n = 399$) by sex, age, diabetes duration, and smoking status. Additionally, controls were required to have a diabetes duration of ≥15 years and an AER within the normal range. All 800 participants were of European ancestry and Finnish residents. The FinnDiane study protocol was approved by the ethics committee of the Helsinki and Uusimaa Hospital District (HUS) (491/E5/2006, 238/13/03/00/2015, and HUS-3313-2018, July 3rd, 2019), and the study was performed in accordance with the Declaration of Helsinki.

### Prospective data collection
Additionally, we gathered follow-up data on the development of kidney failure (dialysis or kidney transplant) for the 401 FinnDiane case participants until December 31, 2017, or death. Altogether 99% (397/401) had data on kidney failure available from the prospective FinnDiane visits or in the Finnish Care Register for Health Care. During the median follow-up time of 7.2 years (interquartile range: 2.9–14.0 years), 196 cases with macroalbuminuria progressed to kidney failure. These data were used to further analyse the top findings from the cross-sectional study of the UK-ROI and FinnDiane.

### Laboratory methodology
All FinnDiane DNA samples were transferred to Belfast for simultaneous laboratory analyses of both cohorts to minimise batch effects. The EZ Zymo Methylation Kit (Zymo Research, USA) was used to bisulphite treat DNA from all participants following overnight incubation (https://files.zymoresearch.com/protocols/_d5001_d5002_ez_ dna_methylationga_o_kit.pdf). All samples were prepared and analysed using the Infinium MethylationEPIC Kit and BeadChips (Illumina, USA) with no protocol deviations. All samples were processed in a consistent laboratory workstream by the same members of trained staff. Methylation arrays were scanned using a dedicated iScan machine with regular monitoring of laser intensity levels. Case and control samples were randomly distributed across the BeadChip arrays. In total, 865,918 sites were examined by the Infinium MethylationEPIC array.

### Data and quality control
Each resulting.idat file generated from the iScan was assessed using Illumina's BACR Software (v1.1.0) for initial QC. This software assessed the data in line with pre-set standards and evaluated hybridisation, extension, dye specificity and bisulphite conversion. An additional QC measure to determine the concordance of average β values generated for seven duplicate samples was completed using the methylation module of Illuminas GenomeStudio (v1.8). GenomeStudio (v1.8) was also used to perform a sex check on the data for all individuals.

Proportional WCCs were estimated using the Houseman method[12], the *minfi* Bioconductor (v3.10) package and the raw.idat files. Estimation of six WCCs was performed using the *estimateCellCounts* function. QC, pre-processing and differential methylation analyses were undertaken in the R statistical environment utilising RnBeads (v2.6.0)[56]. Cross-reactive probes and those located within three base pairs of common SNPs were excluded due to their abilities to map to multiple areas of the genome and affect probe hybridisation, respectively. Unreliable probes and samples were removed using the Greedycut algorithm ($p < 0.05$). Those located on sex chromosomes were also removed. Methylation β values were generated for all CpG sites and normalised using the beta-mixture quantile (bmiq) normalisation method. *M*-values were calculated from the normalised beta-values (*M*-values $=\log_2$(Methylation β value/(1− Methylation β value)).

All software was used following the developer's instructions and QC was completed separately for the UK-ROI and FinnDiane data.

### Epigenome-wide methylation analysis
Analysis of genome-wide methylation was performed separately in both cohorts. *P*-values for differences in CpG methylation levels between DKD cases and controls were computed using the RnBeads (v2.6.0), adapts the Linear Models for Microarray Data (Limma) method[57] for use with methylation data. Aggregation and comparison of DNA methylation levels across four additional genomic regions of interest (CpG islands, genes, gene promoters and genomic tiling) were also completed by RnBeads[56]. In RnBeads, the region-based analyses are calculated as the average difference in means across all CpGs in a region of the two groups being compared (here DKD vs controls) and a combined *p*-value is calculated from all CpGs *p*-values in the region[58].

Three different adjustment models were considered; the Minimal Model, which adjusted for age, sex and the six WCCs; the Minimal Model plus current smoking status, which further adjusted for smoking status at the time of blood sample collection; and the Maximally adjusted Model, which further included HbA$_{1c}$, HDL cholesterol, triglycerides, diabetes duration and BMI. These were completed for the UK-ROI, and FinnDiane data independently and only included individuals with data available for all variables ($n = 1302$ for minimal model; $n = 1175$ for minimal model plus current smoking status; $n = 957$ for maximal model). Summary data are available in Table 1.

### Identification of differentially methylated CpG in DKD
Output data following the QC and RnBeads analyses pertaining to the individual CpG sites, CpG islands, genes, promoters and tiling were shared between both cohorts, and a sample-size weighted meta-analysis was performed using METAL[13]. Specifically, we used the FDR-adjusted *p*-values, the difference in the methylation

means between the case and control groups (to determine the direction of effect on methylation), and the number of individuals (weights) in the sample-size weighted meta-analysis. We considered the commonly used p-value threshold of $p \leq 9.9 \times 10^{-8}$ for epigenome-wide significance to identify epigenome-wide significant associations[25,59]. In addition, we required differentially methylated CpG sites, islands, genes, and promoters to affect methylation levels in the same direction in both cohorts and reach epigenome-wide significance in at least one of the three adjusted models. Differentially methylated CpGs identified with $p \leq 9.9 \times 10^{-8}$ were reported with CpG locations mapped to Human Genome build 37 and annotated using the Illumina Infinium MethylationEPIC v1.0 B4 Manifest File. Manhattan and Quantile-quantile (QQ) plots were drawn following meta-analysis for each adjusted model, using the R package 'qqman' (v0.1.8). Volcano plots were drawn using the R package 'ggplot2' (v3.3.2).

## Differentially methylated CpGs and survival analysis for DKD progression
We extracted individual methylation $M$-values for the top differentially methylated CpGs (corrected $p \leq 9.9 \times 10^{-8}$) from the FinnDiane cohort case participants ($n = 401$). The progression from macroalbuminuria to kidney failure ($n = 196$ events) was studied using Cox proportional-hazards model in the R package 'survival' (v3.2-13) with kidney failure as an outcome, methylation $M$-values as the predictor and with adjustment for age, sex and the six estimated WCCs. Bonferroni correction was applied for the association p-values (0.05/32 CpGs). Altogether 388 individuals had complete data on eight clinical variables (age, sex, age at diabetes onset, systolic blood pressure, HbA$_{1c}$, triglycerides, smoking, retinal photocoagulation). We calculated the concordance statistics (C-index) for Cox proportional-hazards regression model with clinical variables and the six estimated WCCs and for a model with CpG methylation, clinical variables, and the six estimated WCCs using the concordance function available in R package 'survcomp' (v1.44.1). Survival analyses were conducted using R v4.1.3.

## Differentially methylated CpGs and look up in previous EWAS
We conducted a lookup for our top differentially methylated CpGs ($p \leq 9.9 \times 10^{-8}$) in a human whole-blood epigenome-wide association study on DKD performed in 473 individuals with diabetes from the CRIC cohort[10], available at https://susztaklab.com/mqtl/mwas.php.

## Analyses of differentially methylated CpGs to assess potential overlaps with TFs
Differentially methylated CpGs ($p \leq 9.9 \times 10^{-8}$) were additionally individually examined using the eFORGE-TF database[15] available on https://eforge-tf.altiusinstitute.org/, which allowed each CpG site to be assessed for transcription factor motif enrichment using previously acquired data. The 'fkidney' dataset, formed from seven experiments, was used for this assessment. All transcription factors overlapping the probe binding site gaining FIMO $p < 1 \times 10^{-5}$ were reported.

## Enrichment analyses on gene ontology, pathways, and traits
Genetic ontology (GO) and KEGG enrichment analyses were performed using the *gometh* function in the R (v4.0) using R package 'missmethyl' (v1.28.0)[60]. Specifically, the hypergeometric is used for testing genomic locations (relation to gene and relation to CpG island) for enrichment or depletion in missmethyl. Enrichment analyses for genomic locations and traits were performed within the web-based EWAS toolkit (https://ngdc.cncb.ac.cn/ewas/toolkit)[61]. In the EWAS toolkit, weighted Fisher's exact test is used for calculating the probability of co-occurrence between differentially methylated CpGs in this study and differentially methylated CpGs for other traits in the EWAS Atlas.

## Assessment of gene expression profiles utilising additional datasets
We individually searched genes that contained significant differentially methylated CpGs ($p \leq 9.9 \times 10^{-8}$) in Nephroseq v4 (www.nephroseq.org), a database containing gene expression profiles from previous kidney disease studies. We searched studies assessing gene expression in DKD kidneys (vs controls) by selecting the 'Group' filter (under 'primary filters') as 'Diabetic Nephropathy', which resulted in seven datasets. For each gene, we reported all significant gene expression changes ($p \leq 0.05$, fold change of at least ±1.5) from the 'Disease vs Control' analysis, which were displayed by ordering by 'Over-Expression: P-Value' and 'Under-Expression: P-Value'. The resulting seven datasets included two datasets from Woroniecka et al.[21], including 22 glomerular (nine DKD cases) and 22 tubular (ten DKD cases) kidney samples collected from healthy, living transplant donors and diagnostic kidney biopsies. Two datasets from Schmid et al. comprising tubulointerstitial samples of 24 kidney biopsies (13 with histological evidence of DKD and 11 histologically normal kidney biopsies from patients with minimal change disease or healthy controls). Two datasets from Ju et al.[20], including kidney biopsy data microdissected into glomerular and tubule-interstitial compartments from 12 individuals with DKD, which were compared to healthy living donors or other diseases. One dataset from the European Renal cDNA Bank-Kroener-Fresenius biopsy bank (ERCB) that included tubulointerstitial kidney samples from ten individuals with DKD samples and nine healthy living donor samples.

The same gene list was also interrogated for altered gene expression in kidney biopsy samples for which RNA-Seq data were available from the North Dublin Renal Biobank. The North Dublin Renal Biobank (NDRBB) study protocol was approved by Beaumont Hospital Ethics Committee. The kidney biopsy study was approved by the Institutional Review Board of the National Institute of Diabetes and Digestive and Kidney Diseases, and each participant signed an informed consent document. There was no financial compensation. RNA-seq data were generated from whole-kidney biopsy tissue from individuals ($n = 44$, mean age 49 years, 26 males/18 females) with multiple CKD aetiologies. A linear regression model was used to examine the correlation between normalised transcript counts and clinico-pathological variables of eGFR and %TIF, with age and sex used as co-variables. Correlation analyses were performed using the *cor.test* function in base R (v4.0.3). Differentially expressed transcripts in progressive versus stable CKD patients were identified using the R package 'Limma' (v3.46.0), following adjustment for age and sex. FDR adjustment was applied using the R package 'Multtest' (v2.46.0) and an FDR cut-off of $p < 0.05$ was deemed statistically significant.

The same set of genes were additionally examined for altered gene expression in a previous study conducted on samples from individuals with T2D and known renal status. Participants included in this investigation were Pima Indians with T2D[62] using kidney biopsy samples ($n = 97$, mean age 47.3 years, 24 males/73 females). The study was approved by the Institutional Review Board of the National Institute of Diabetes and Digestive and Kidney Diseases. All participants provided informed consent and there was no financial compensation. Expression profiling of the kidney biopsies was carried out using Affymetrix GeneChips (HumanGenome U133 Array and U133Plus2 Array and Affymetrix Human Gene ST GeneChip 2.1)[63,64], and RNA-seq (Illumina)[65]. For morphometry measures, kidney biopsy tissue was prepared for light and electron microscopy studies according to standard procedures[66–68]. The following glomerular structural parameters were measured by unbiased morphometry on electron microscopy images as described elsewhere[66,67,69]: glomerular basement membrane width[70,71], percentage of intact foot processes on both the peripheral and mesangial glomerular basement membrane[72], numerical density of podocyte cell per glomerulus[73], percentage of endothelial fenestration falling on the peripheral glomerular basement

membrane[74], foot process width in peripheral glomerular basement membrane[72] (surface volume of peripheral glomerular basement membrane per glomerulus[70,71], mesangial fractional volume[70,71]; and the fractional podocyte volume per glomerulus[75]. Correlation coefficients were calculated using the spearman correlation method implemented in GraphPad Prism v8.0 (GraphPad Software).

## Mendelian randomisation

Two-sample Mendelian randomisation was used to assess potential causality between methylation levels at identified CpGs and DKD. For the genetic variant-exposure associations, we used whole-blood mQTLs, which are single nucleotide variants with a known impact on the methylation level at a specific CpG site. We searched mQTL variants for each CpG from results from the Genetics of DNA Methylation Consortium (GoDMC)[23], and selected both cis and trans meQTLs available on the GoDMC site, but only those that were independently associated ($p < 10^{-5}$, $r^2 > 8.0$) with methylation at each CpG site (mqtldb.godmc.org.uk). We retrieved the genetic variant-outcome associations from the latest genome-wide association study on DKD in T1D[65]. Causal ORs were calculated for each CpG using the Wald ratio test (one mQTL) or the inverse variance weighted method (mQTL > 1). If several mQTLs ($n > 2$) were identified for the methylation levels at a CpG, we performed additional tests (heterogeneity and the Egger intercept test) to address potential pleiotropy, which could violate the instrumental variable assumptions. All Mendelian randomisation analyses were conducted in R (v 4.1) using R package 'TwoSampleMR' (v 0.5.6).

A figure summarising all methods is included in Supplementary Fig. 1.

## Reporting summary

Further information on research design is available in the Nature Portfolio Reporting Summary linked to this article.

## Data availability

The summary data generated in this study (all three models) have been deposited at GENIE Updates | RenGenPECT (qub.ac.uk). Individual-level data for the study participants (participant-level genome-wide CpG methylation and clinical data for participants in UK-ROI and FinnDiane and gene expression data in American Indians with T2D) are not publicly available due to ethical and legal reasons and due to the consent provided by the participant at the time of data collection. Access, which is subject to local regulations, can be obtained upon reasonable request by contacting the following persons (FinnDiane study: niina.sandholm@helsinki.fi, UK-ROI: a.j.mcknight@qub.ac.uk, American Indian population of Pima Indians with T2D; vijin@med.u-mich.edu). Upon approval, analyses need to be performed on a local server (with protected, user-specific access) and requires signing non-disclosure and privacy agreements. The summary data used for the two-sample Mendelian randomization analyses are available on the GoDMC site (SNP-CpG associations; http://mqtldb.godmc.org.uk/) and on the type 2 diabetes knowledge portal (SNP-DKD associations; https://t2d.hugeamp.org/dinspector.html?dataset=GWAS_DNCRI). The data from the eforge TF database can be accessed on https://eforge-tf.altiusinstitute.org/. The nephroseq v4, containing the seven DKD kidney gene expression datasets (four cohorts), is a free platform to the academic and non-profit community and data deposited there can be analysed and accessed after registration and login (www.nephroseq.org). The nephroseq datasets 'Woroniecka Diabetes TubInt' ($n = 2$) are also available under accession code http://www.ncbi.nlm.nih.gov/geo/query/acc.cgi?acc=GSE30122. The Northern Dublin Renal Biobank RNAseq data are available under accession code https://www.ncbi.nlm.nih.gov/geo/query/acc.cgi?acc=GSE137570. For the SNP-DKD associations the dataset 'late diabetic kidney disease (diabetic nephropathy)' was used. The eQTM data searched in this study is available on http://www.susztaklab.com/Kidney_meQTL/eQTM.php (kidney eQTM), https://static-content.springer.com/esm/art%3A10.1186%2Fs13148-021-01041-5/MediaObjects/13148_2021_1041_MOESM2_ESM.xlsx (whole blood eQTM, Framigham study), http://bbmri.researchlumc.nl/atlas/#query (whole blood eQTM, BIOS consortium) and https://static-content.springer.com/esm/art%3A10.1186%2Fs12864-018-4842-3/MediaObjects/12864_2018_4842_MOESM2_ESM.txt (monocytes, eQTM).

## Code availability

Any custom code has been uploaded to https://github.com/EmmDah/EWAS-Meta-analysis-on-DKD-in-T1D.

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

## Acknowledgements

The authors are grateful to the late Carol Forsblom (1964–2022), the international coordinator of the FinnDiane Study Group, for his substantial contribution. We thank all individuals who participated in this study and acknowledge all physicians and nurses who were involved in their recruitment. We would like to acknowledge the GENIE consortium (Supplementary Table 10) and Warren 3 and Genetics of Kidneys in Diabetes (GoKinD) Study Group. The GENIE_UK data include samples recruited as part of the Warren3/U.K. GoKinD Study Group, jointly funded by Diabetes UK and the Juvenile Diabetes Research Foundation and includes the following individuals: Professor A.P. Maxwell, Professor A.J. McKnight, Dr. D.A. Savage (Belfast); Dr. J. Walker (Edinburgh); Dr. S. Thomas, Professor G.C. Viberti (London); Professor A.J.M. Boulton (Manchester); Professor S. Marshall (Newcastle); Professor A.G. Demaine and Dr. B.A. Millward Plymouth); and Professor S.C. Bain (Swansea). We would like to acknowledge Dr Katalin Susztak and Dr Hongbo Liu for offering advice regarding the CRIC study lookups. The FinnDiane study was supported by grants from the Folkhälsan Research Foundation, the Wilhelm and Else Stockmann Foundation, the Liv och Hälsa Foundation, Helsinki University Central Hospital Research Funds (EVO TYH2018207), the Novo Nordisk Foundation (NNF OC0013659), the Sigrid Juselius Foundation, and Academy of Finland (299200 and 316664). The GEnetics of Nephropathy an International Effort (GENIE) is supported through the Medical Research Council (MC_PC_15025), Public Health Agency R&D Division (STL/4760/13), Science Foundation Ireland (SFI15/US/B3130), NIH R01_DK081923, R01_DK105154 and R01_DK132299. LJS was the recipient of a Northern Ireland Kidney Research Fund Fellowship and is supported by an award from HSC R&D Division STL/5569/19 and UKRI MRC MC_PC_20026. L.J.S., K.A., A.P.M. and A.J.M. are also supported by Science Foundation Ireland and the Department for the Economy, Northern Ireland partnership award 15/IA/3152. EB is a University College Dublin Ad Astra fellow. This study was also supported, in part, by funding from NIH/NIDDK through the George M. O'Brien Michigan Kidney Translational Core Center, grant 2P30-DK-081943, and by the Intramural Research Program of NIDDK.

## Author contributions

L.J.S. and E.D. designed the study, generated new data, performed the analyses, contributed to data interpretation, drafted the manuscript, and approved the submitted version. E.D. performed the Mendelian Randomization. A.S., N.S., C.F., A.P.M., P.H.G and A.J.M. conceived and designed the study, contributed to the interpretation of data, iteratively critically revised the manuscript for important intellectual content, and approved the submitted version. Additionally, A.S. performed the prospective analysis. K.K., J.K., C.W. and K.A. generated the methylation data, contributed to the interpretation of data, revised the manuscript, and approved the submitted version. R.D., E.B., V.N., D.F., R.G.N., H.C.L., G.J.M., D.A., P.J.C., J.C., R.M.S., M.K., J.N.H., C.G., P.C., D.S. and J.C.F. contributed to the interpretation of the results, revision of the manuscript, and approved the submitted version.

## Competing interests

P.-H.G has received investigator-initiated research grants from Eli Lilly and Roche, is an advisory board member for AbbVie, Astellas, AstraZeneca, Boehringer Ingelheim, Cebix, Eli Lilly, Janssen, Medscape, Merck Sharp & Dohme, Mundipharma, Nestle, Novartis, Novo Nordisk and Sanofi; and has received lecture fees from AstraZeneca, Boehringer Ingelheim, Eli Lilly, Elo Water, Genzyme, Merck Sharp & Dohme, Medscape, Novartis, Novo Nordisk, PeerVoice and Sanofi. J.C.F. has received consulting honoraria from Goldfinch Bio and AstraZeneca, and speaker fees from Novo Nordisk, AstraZeneca, and Merck for research lectures over which the author had full control of content. J.N.H. hold equity in Camp4 Therapeutics. Dr. M. Kretzler reports grants and contracts outside the submitted work through the University of Michigan with NIH, Chan Zuckerberg Initiative, JDRF, AstraZeneca, NovoNordisk, Eli Lilly, Gilead, Goldfinch Bio, Janssen, Boehringer-Ingelheim, Moderna, European Union Innovative Medicine Initiative, Certa, Chinook, amfAR, Angion, RenalytixAI, Travere, Regeneron, IONIS; consulting fees through the University of Michigan from Astellas, Poxel and Janssen; and a patent PCT/EP2014/073413 "Biomarkers and methods for progression prediction for chronic kidney disease" licensed. R.M.S. receives funding from Travere Therapeutics. The funders listed had no role in any part of this study (design, data collection, management, analysis, interpretation, preparation, review or approval of the manuscript). The remaining authors declare no competing interests.

## Additional information

[1]Molecular Epidemiology Research Group, Centre for Public Health, Queen's University Belfast, Belfast, UK. [2]Folkhälsan Institute of Genetics, Folkhälsan Research Center, Helsinki, Finland. [3]Department of Nephrology, University of Helsinki and Helsinki University Hospital, Helsinki, Finland. [4]Research Program for Clinical and Molecular Metabolism, Faculty of Medicine, University of Helsinki, 00290 Helsinki, Finland. [5]Diabetes Complications Research Centre, Conway Institute, School of Medicine, University College Dublin, Dublin, Ireland. [6]Department of Medicine-Nephrology, University of Michigan School of Medicine, Ann Arbor, MI 48109, USA. [7]Department of Pediatrics-Nephrology, University of Michigan School of Medicine, Ann Arbor, MI 48109, USA. [8]Chronic Kidney Disease Section, National Institute of Diabetes and Digestive and Kidney Diseases, Phoenix, AZ, USA. [9]Programs in Metabolism and Medical & Population Genetics, Broad Institute, Cambridge, MA, USA. [10]Diabetes Unit and Center for Genomic Medicine, Massachusetts General Hospital, Boston, MA, USA. [11]Herbert Wertheim School of Public Health and Human Longevity Science, University of California San Diego, La Jolla, CA, USA. [12]Department of Nephrology and Transplantation, Beaumont Hospital and Department of Medicine Royal College of Surgeons in Ireland, Dublin 9, Ireland. [13]Department of Internal Medicine, University of Michigan, Ann Arbor, Michigan, USA. [14]Division of Endocrinology, Boston Children's Hospital, Boston, MA, USA. [15]Department of Pediatrics and Genetics, Harvard Medical School, Boston, MA, USA. [16]Mater Misericordiae Hospital, D07 K201 Dublin, Ireland. [17]Department of Medicine, Harvard Medical School, Boston, MA, USA. [18]Regional Nephrology Unit, Belfast City Hospital, Belfast, Northern Ireland, UK. [19]Department of Diabetes, Central Clinical School, Monash University, Melbourne, Victoria, Australia. [20]These authors contributed equally: Laura J. Smyth, Emma H. Dahlström.
✉e-mail: niina.sandholm@helsinki.fi; a.j.mcknight@qub.ac.uk

## GENIE consortium

Laura J. Smyth ⬓[1,20], Emma H. Dahlström ⬓[2,3,4,20], Anna Syreeni ⬓[2,3,4], Katie Kerr[1], Jill Kilner ⬓[1], Ross Doyle[5], Eoin Brennan ⬓[5], Viji Nair[6], Damian Fermin[7], Christopher Wooster ⬓[1], Darrell Andrews[5], Kerry Anderson[1], Gareth J. McKay[1], Joanne B. Cole ⬓[9,10], Rany M. Salem ⬓[11], Matthias Kretzler ⬓[13], Joel N. Hirschhorn[9,14,15], Denise Sadlier[16], Catherine Godson[5], Jose C. Florez ⬓[9,10,17], Carol Forsblom ⬓[2,3,4], Alexander P. Maxwell ⬓[1,18], Per-Henrik Groop ⬓[2,3,4,19], Niina Sandholm ⬓[2,3,4] ✉ & Amy Jayne McKnight ⬓[1] ✉

A full list of members and their affiliations appears in the Supplementary Information.

