## [Peer Review File · Nature Communications]

Epigenome-wide meta-analysis identifies DNA methylation biomarkers associated with diabetic kidney diseaseREVIEWER COMMENTS

Reviewer #1 (Remarks to the Author):

In this study, the authors have compared methylation status in DNA obtained from blood in patients with diabetes and established albuminuria, taken as a surrogate for underlying diabetic kidney disease, and controls without albuminuria despite 15 or more years of diabetes. The samples are derived from two extant cohorts, analysed together. These are the UK ROI and Finn Diane cohorts, in aggregate giving samples from 1,304 patients, from which 32 CpGs were identified in 28 different loci. Subsequent analyses are presented that evaluate the effects of other differences observed between cases and controls in clinically important parameters, correlate the data to that previously presented in another cohort (the CRIC study cohort), link the CpGs to genes on the basis of proximity and evaluate differential expression of identified genes in published datasets from kidney biopsy tissue, evaluate gene ontology and underlying pathways for the loci, and seek evidence for causality of identified associations using Mendelian Randomisation. Overall, this is an important study that adds significantly to prior findings by these and other investigators. I have points for the authors to consider:

Major points

1. The authors should comment on the inherent limitation of making inferences about epigenetic control in kidney cells based on findings in blood DNA, including:
 - i. The highly specialised and varied phenotype of cells in the kidney, and the potential for important epigenetic mechanisms to be cell-specific
 - ii. That the timescale of epigenetic alterations driven by environment may depend on the lifespan of the cell. Peripheral leucocyte lifespan is very different from, for example, podocyte lifespan (podocytes being a non-renewing cell population across the lifespan, and a cell with critical roles in DKD pathogenesis). Related, in DKD, key risk factors may exert their influence many years after the effect (for example the effects of relatively short term intensive glycaemic control seen many years later in DKD risk in the seminal Diabetes Control and Complications Study).
2. There is an existing literature on similar epigenetics studies in DKD, appropriately cited here by the authors. The current study is the largest by a significant margin. There has been some comparative analysis in the current paper with this previous work, but I was left unclear having read this paper what the overall level of agreement of the current submission is with this previous work. Could the authors present a table or similar to detail this?
3. Important factors appear different between cases and controls, including smoking history, glycated haemoglobin, lipid parameters, BMI and diabetes duration. Adjusting for these reduced the number of significant CpG's to 7 -ie. A substantial proportion of the epigenetic association with DKD was explained when these factors were considered. It is possible then that the epigenetic markers identified are causal for the risk of DKD induced by these clinical parameters, but equally these associations may be epiphenomena.
4. I have several related points concerning the studies of expression data in kidney tissue. There is a tabulation of findings from the data repository "nephroseq" together with secondary analysis of RNAseq data. There is additionally description of evaluation depending on a prior morphometry study using kidney biopsy sections from Pima Indians.
 - i. Overall, the confidence of the linkage of epigenetic markers to altered gene expression in kidney would be substantially improved by presentation of additional quantitative experimental data demonstrating this.
 - ii. It is hard to pick out the quantitative findings from the data as presented. The table lists genes as "increased" or "decreased" and then the RNAseq data is presented in the body of the text but not the main figures. Presentation of correlation to "various morphology parameters" is not sufficient detail on

the morphometry results.

iii. I found it hard to discern additional key points, including the statistical approaches and whether these were adjusted for multiple testing, and where analysis was whole kidney versus glomerular, tubular, and other subcompartments, and how these had been discriminated in the analysis.

iv. The North Dublin Renal biobank data were adjusted for age and sex, but not apparently for the other clinical parameters corrected for in the epigenetic modelling

Minor points

5. Can the authors comment on the choice of a combined analytic approach with an evaluation of "same direction of effect" in the two cohorts, rather than a discovery and validation cohort approach?

6. Can the authors present an analysis that demonstrates the additional discriminatory/predictive power of adding analysis of methylation status to existing clinical risk prediction algorithms (based on eGFR and proteinuria) in predicting eGFR decline?

7. Did the authors consider censoring in the analysis of the 397 cases for whom follow up data was available? Associations with cardiovascular risk and hence death, and unavailability of follow up data may have been a source of bias.

8. In the absence of new data, the subtitle for kidney tissue analysis (line 203) should make it clear that secondary analysis rather than new primary data are presented.

Reviewer #2 (Remarks to the Author in attachment):

Epigenome-wide meta-analysis identifies DNA methylation biomarkers associated with diabetic kidney disease

The study reports findings from an epigenomic-wide association study of type 1 diabetes kidney traits (DKD) using whole blood DNA methylation data from two cohorts: the United Kingdom-Republic of Ireland (UK-ROI) and Finland (FinnDiane) for a total of 1,304 participants (651 cases and 653 controls with and without DKD). Main findings are the association of DKD with 32 CpGs (31 dmCpGs from a minimally adjusted model and an additional CPG identified when adjusting for smoking). Of the 32 CpGs, 21 were also associated with kidney failure over time in the FinnDiane study. Replication was performed through a query of summary statistics from the CRIC study of individuals with type 2 diabetes. Several analyses to assess regulatory function were performed using datasets for transcription factors and gene expression in kidney tissue/cells, and a Mendelian randomization identified evidence for causal effects for a CpG within the REV1 gene.

This is one of the largest EWAS of DKD in type 1 individuals of European ancestry and therefore it provides novel findings for this trait. However, the samples used for replication are likely not comparable to the discovery given CRIC participants have type 2 diabetes and the definition of DKD seems different. Given the difficulties of finding replication samples for type 1 diabetes, I suggest that the authors discuss the replication limitations in the discussion. This is important for future studies as these CpGs still need to be replicated.

The definition of DKD for the discovery sample needs to be clarified. The main text defines DKD cases as persistent macroalbuminuria, i.e. albumin 119 excretion rate (AER) > 300 mg/ml in urine), and controls had an AER within the normal range despite 120 a long duration of T1DM (≥ 15 years). However, the methods describe DKD as a low eGFR. Given replication was done for eGFR traits in CRIC, this needs to be clarified.

Related to the analyses, the case-control studies were matched on age, sex and other variables, so conditional analyses would be the more appropriate approach. The longitudinal analysis methods are not described, included the covariates used in models.

Clarify why principal components were not used in EWAS models. It seems that the author used Methylation M-values for analysis, but it is not clear if methylation was used as a predictor or as an outcome.

Methods used for differential methylation levels within regions and gene-based analysis described in results in the paragraph starting at line 257 are not described in detail in methods.

The authors need to report lambdas for EWAS. Figure 1 C, qq plot distribution needs some comment.

Mendelian randomization results do not include analysis to verify assumptions.

Table 1, add eGFR values for the studies. Table 2, add footnote that the CRIC study has participants with T2D CKD

What are the differences in results reported in Figure 1 and Figure 3?

The data sharing section is missing.

The authors could state how their results move the field forward and their view of future studies.

Dear Reviewers,

Thank you for the careful evaluation of our manuscript, and for your helpful comments. We have responded to each comment in the point-by-point response below, and revised the manuscript based on your feedback (with changes highlighted in yellow). We hope that the changes will address your concerns and that the raised questions have been satisfactorily answered.

Sincerely,

Dahlström, Smyth, Sandholm and McKnight, on behalf of all authors

REVIEWER COMMENTS

Reviewer #1 (Remarks to the Author):

In this study, the authors have compared methylation status in DNA obtained from blood in patients with diabetes and established albuminuria, taken as a surrogate for underlying diabetic kidney disease, and controls without albuminuria despite 15 or more years of diabetes. The samples are derived from two extant cohorts, analysed together. These are the UK ROI and Finn Diane cohorts, in aggregate giving samples from 1,304 patients, from which 32 CpGs were identified in 28 different loci. Subsequent analyses are presented that evaluate the effects of other differences observed between cases and controls in clinically important parameters, correlate the data to that previously presented in another cohort (the CRIC study cohort), link the CpGs to genes on the basis of proximity and evaluate differential expression of identified genes in published datasets from kidney biopsy tissue, evaluate gene ontology and underlying pathways for the loci, and seek evidence for causality of identified associations using Mendelian Randomisation. Overall, this is an important study that adds significantly to prior findings by these and other investigators. I have points for the authors to consider:

Major points interested in

1. The authors should comment on the inherent limitation of making inferences about epigenetic control in kidney cells based on findings in blood DNA, including:

- i. The highly specialised and varied phenotype of cells in the kidney, and the potential for important epigenetic mechanisms to be cell-specific**
- ii. That the timescale of epigenetic alterations driven by environment may depend on the lifespan of the cell. Peripheral leucocyte lifespan is very different from, for example, podocyte lifespan (podocytes being a non-renewing cell population across the lifespan, and a cell with critical roles in DKD pathogenesis). Related, in DKD, key risk factors may exert their influence many years after the effect (for example the effects of relatively short term intensive glycemic control seen many years later in DKD risk in the seminal Diabetes Control and Complications Study).**

Authors reply: Yes, we agree. The use of blood DNA to make inferences about genetic control in kidneys is undeniably a limitation. We also acknowledge the heterogeneity on the lifespan of the cells (blood vs podocytes). We cannot therefore be completely sure that the changes in blood reflect changes in the kidneys. Measuring methylation in kidneys in our study was not possible as we do not have access to kidney samples, as individuals with type 1 diabetes and kidney disease are rarely biopsied. In addition, the correlation between DNA methylation levels across tissues has been reported to be as high as 0.86 (Byun et al, 2009), further supporting that the methylation levels in kidneys could be to some extent reflected in the blood. Importantly, studying the DNA methylation in an accessible tissue such as blood enables identification of biomarkers that could be used in a routine clinical setting or population-based cohort, and this was indeed one of our primary aims. However, this is an important topic and based on the reviewer comment we have now added the following section to the discussion:

Using DNA methylation levels measured from the blood instead of kidneys to make inferences about changes in the kidneys is a potential limitation. The lifespan of the cells in kidneys vs blood is broadly different, which might impact the observed methylation changes. For example, podocytes - the highly specialised cells of the kidney glomerulus that line the outer surface of the glomerular capillaries - are terminally differentiated, whereas the peripheral blood cells are dividing cells with an average lifespan of 120 days. We cannot completely elude whether the changes occurring in peripheral blood reflect the changes that are acquired throughout the lifespan of the podocyte. However, kidney biopsies are invasive procedures that are performed in individuals with type 1 diabetes only if there is a clinical suspicion that a non-diabetic kidney disease is likely; therefore, such samples are rarely available. In addition, methylation levels across tissues in an individual are highly correlated ($r=0.86$, Byun et al.,2009) and we have previously shown that DNA methylation changes observed in blood were reflected in kidney biopsies (Smyth et al.,2014). However, owing to the very specialised and varied phenotype of the cells in the kidneys, there is a possibility that some changes in the methylation levels accompanying diabetic kidney disease are not evident in our study. Future studies will have to characterise these changes and their relevance in diabetic kidney disease. Nevertheless, studying the DNA methylation in an accessible tissue such as blood enables identification of biomarkers that could be used in a routine clinical setting or population-based cohort, where biopsies are not feasible.

References:

Byun, Hyang-Min, et al., Epigenetic profiling of somatic tissues from human autopsy specimens identifies tissue- and individual-specific DNA methylation patterns, *Human Molecular Genetics*, Volume 18, Issue 24, 15 December 2009, Pages 4808–4817, <https://doi.org/10.1093/hmg/ddp445>

Smyth, Laura J et al. "DNA hypermethylation and DNA hypomethylation is present at different loci in chronic kidney disease." *Epigenetics* vol. 9,3 (2014): 366-76. <https://doi:10.4161/epi.27161>

2. There is an existing literature on similar epigenetics studies in DKD, appropriately cited here by the authors. The current study is the largest by a significant margin. There has been some comparative analysis in the current paper with this previous work, but I was left unclear having read this paper what the overall level of agreement of the current submission is with this previous work. Could the authors present a table or similar to detail this?

Authors reply: Thank you for the suggestion! We have now performed additional lookups to explore the agreement of our study with the previous blood-based EWAS studies on DKD (Sheng et al., Smyth et al. and Qiu et al.) and a kidney tubuli-based EWAS on tubular fibrosis in 91 individuals of which half had diabetes (Gluck et al.). We have included DKD-associated dmCpGs from these previous studies and their association

results in our EWAS meta-analysis on DKD (Supplemental Table 11 of the revised manuscript). Some of the previously identified dmCpGs for DKD-related phenotypes (eGFR, albuminuria) are also associated with DKD in our study. For example, the top findings in Sheng et al. 2020 (Blood-based EWAS in the CRIC study including 473 participants with diabetes) and Smyth et al. located in the Zinc finger region are also the dmCpGs with the lowest p-values in our study (cg17944885 and cg25544931). Not surprisingly, some overlap was observed with Smyth et al., 2021 (~9%, 43 CpGs with $p < 0.05$ out of 477 found in our EWAS/minimal model) as we have ~18% of sample overlap with that study. There is also some overlap with the results from the CRIC study (Sheng et al.); 13 (14%) of the eGFR-associated CpGs are also associated with DKD in our study ($p < 0.05$). No overlap is observed with Qiu et al (ESRD in Blood-based EWAS in 181 Pima Indians with type 2 diabetes). Of note, Qiu et al. also observed no overlap with the CRIC study either. Furthermore, as reported in table 2, nine (27%) of the 33 DKD-associated dmCpGs in our study were either associated with eGFR or albuminuria in the CRIC study (Sheng et al).

The lookups have been included in the supplemental table 11 of the revised manuscript.

We have now revised the text to include a section (“Overlap with CpGs detected in previous EWAS studies” with these findings (and with the trait enrichment results in Figure 4c):

To compare overlap with previous epigenome-wide association studies on DKD, we looked up CpGs previously associated with DKD-related phenotypes (ESRD, eGFR, albuminuria) in epigenome-wide association studies performed in the blood (Qiu et al., Sheng et al. Smyth et al.) or kidney tubuli (Gluck et al.) in our meta-analysis. We observed some overlap with results from the CRIC study (Sheng et al.), as 13 (14%) of the eGFR-associated CpGs are also associated with DKD in our study (Supplementary Table 11). Also, 43 dmCpGs that we previously identified for ESRD in type 1 diabetes were associated with DKD in this study. Methylation levels of three CpGs in kidney tubuli (out of 65) associated with interstitial fibrosis were also differentially methylated in our meta-analysis on DKD. No overlap was observed with the findings from Qiu et al for EWAS on ESRD in Pima Indians.

3. Important factors appear different between cases and controls, including smoking history, glycated haemoglobin, lipid parameters, BMI and diabetes duration. Adjusting for these reduced the number of significant CpG's to 7 -ie. A substantial proportion of the epigenetic association with DKD was explained when these factors were considered. It is possible then that the epigenetic markers identified are causal for the risk of DKD induced by these clinical parameters, but equally these associations may be epiphenomena.

Authors reply: Adjusting for all these confounders indeed reduced the number of EWAS significant CpGs to only seven. However, the adjustment did not completely remove the significance of these sites. For several sites, the significance level dropped to just under the EWAS significance threshold. Adjusting for additional covariates also resulted in a lower number of participants (1,302 participants in the minimal model vs 957 participants in the maximal model). It is, therefore, possible that the epigenome-significance of some of the sites was lost due to the reduced power. However, another explanation may very well be that the effect of these dmCpGs on DKD is mediated by intermediate clinical parameters. Given that we have only seven instrumental variables (meQTLs; SNPs affecting methylation levels at the CpGs), a mediation analysis would be very limited. However, this we have highlighted in our conclusion:

“More comprehensive meQTL data are needed to allow more systematic MR analyses for causality as well as functional studies assessing the biological underpinnings of these loci.”

The trait enrichment analyses in Figure 4, also highlight the enrichment of associations with other traits for the DKD-associated dmCpGs. These traits include for example BMI, smoking and blood pressure which are

identified as enriched associations for the DKD-associated CpGs. We have clarified this in the results of the text (section “Overlap with CpGs detected in previous EWAS studies”):

“ To further evaluate the overlap with other traits, we performed trait enrichment analysis for DKD-associated CpGs ($p < 1 \times 10^{-5}$) against the reported associations in the EWAS atlas. The top 20 most significantly associated traits are shown in Figure 4c. The DKD-associated CpGs showed significant overlap with several established risk factors for DKD, such as ageing, blood pressure, eGFR and smoking.”

4. I have several related points concerning the studies of expression data in kidney tissue. There is a tabulation of findings from the data repository “nephroseq” together with secondary analysis of RNAseq data. There is additionally description of evaluation depending on a prior morphometry study using kidney biopsy sections from Pima Indians.

i. Overall, the confidence of the linkage of epigenetic markers to altered gene expression in kidney would be substantially improved by presentation of additional quantitative experimental data demonstrating this.

Authors reply: We agree; linking epigenetic markers, i.e. methylation, to altered gene expression requires eQTM data (associations between the methylation levels at a specific site and gene expression). There is eQTM data from kidneys recently published (Liu et al., 2022) and we used these data to understand whether the DKD-associated CpGs identified in blood in this study are also connected to altered gene expression in the kidneys. Of the 33 differentially methylated CpGs in the blood of individuals with DKD, methylation at 10 CpGs was also associated with altered gene expression in the kidneys in cis (also known as eQTMs, Supplementary Table 7). Furthermore, seven of the CpGs that were eQTMs, were associated with the expression of the gene that they were located within (Table 3). We also used blood eQTM data as a source to see whether the kidney eQTMs are in line with the whole blood eQTMs. However, for only two sites, we had overlapping eQTM data (TXNIP and CPT1A CpGs) for blood and kidney (Table 3). Nevertheless, the kidney eQTM and whole blood eQTM results for these were in agreement. This finding is also in line with a recent preprint suggesting that effect directions of eQTMs are consistent across tissues (Shueang et al, Bioarxiv).

References:

Liu, H., Doke, T., Guo, D. et al. Epigenomic and transcriptomic analyses define core cell types, genes and targetable mechanisms for kidney disease. *Nat Genet* 54, 950–962 (2022). <https://doi.org/10.1038/s41588-022-01097-w>

Shuang Li, et al. Integration of public DNA methylation and expression networks via eQTMs improves prediction of functional gene–gene associations. *bioRxiv* 2021.12.17.473125; doi: <https://doi.org/10.1101/2021.12.17.473125>

ii. It is hard to pick out the quantitative findings from the data as presented. The table lists genes as “increased” or “decreased” and then the RNAseq data is presented in the body of the text but not the main figures. Presentation of correlation to “various morphology parameters” is not sufficient detail on the morphometry results.

Authors reply: We apologize for being unclear here; we don’t have RNA seq data in the primary cohort (FinnDiane/UK-ROI), therefore we have not presented that data in the main figures. However, external cohorts were used to check gene expression in kidney tissues from biopsies. We have now modified our table to make it clearer; for the eQTM results we have explicitly stated whether the DKD-associated

differentially methylated CpG identified in our study is also a known eQTM and how the methylation influence the expression (increase/decrease) of the listed gene in table 3. We have also added the correlation coefficients to table 3 for the morphology parameters and the fold changes for the nephroseq data, as well as the log₂ ratios and correlations in the Northern Dublin renal bank (NDRB) dataset.

iii. I found it hard to discern additional key points, including the statistical approaches and whether these were adjusted for multiple testing, and where analysis was whole kidney versus glomerular, tubular, and other sub compartments, and how these had been discriminated in the analysis.

Authors reply: Thank you for noticing this unclarity; we did not adjust the gene expression differences for multiple testing (except for the NDRB dataset as these results were explicitly calculated for this study). The information on gene expression in biopsies was primarily included to provide supporting (functional/biological) evidence for the genes containing dmCpGs and their involvement in kidney disease, particularly diabetic kidney disease. Therefore, we reported all the gene expression results with a $p < 0.05$ in table 3. This has been added to the table 3 description:

“Table 3. Supporting evidence from eQTM data (whole-blood, monocytes, and kidney) and gene expression data from kidney biopsies for the genes containing DKD-associated CpGs (Associations with $p < 0.05$ reported)”

We have also added a note below the table stating that NDRB results were FDR adjusted:

“The Northern Dublin Renal Bank (NDRB) included whole-kidney biopsies from 44 individuals with multiple CKD etiologies. Reported correlations and log₂-ratios have an FDR-adjusted p -value < 0.05 .”

The statistical approaches for the nephroseq data are dependent on the dataset, and as we used seven datasets from the nephroseq database (and as these are lookups based on mainly published datasets), we did not include the statistical approaches for these studies. We have now revised the description of the datasets in table 3 and added the fold changes to the table 3 for each gene:

“Gene expression in DKD kidneys (FC vs. non-DKD, Nephroseq)”

We also realized that we had not described the statistical methods for the Northern Dublin Renal Bank (NDRB), so that has now added the following to the methods section:

“The same gene list was also interrogated for altered gene expression in kidney biopsy samples for which RNA-Seq data were available from the North Dublin Renal Biobank (NDRB). Here, RNA-seq data were generated from whole kidney biopsy tissue from individuals ($n=44$) with multiple CKD aetiologies. A linear regression model was used to examine the correlation between normalized transcript counts and clinico-pathological variables of eGFR and %TIF, with age and sex used as co-variables. Correlation analyses were performed using the `cor.test` function in base R. Differentially expressed transcripts in progressive versus stable CKD patients were identified using the `limma` package in R, following adjustment for age and sex. An FDR cut-off of $p < 0.05$ was deemed statistically significant. RNA-seq data are deposited at the Gene Expression Omnibus (GSE137570).”

In that dataset, RNA-seq data were generated from RNA extracted from renal biopsy (i.e. whole kidney, not microdissected into glomerular or tubular compartments), and this information has now also been added to the table. Table 3 now includes the information whether the analysis concerned whole kidney or only glomerular or tubular compartments. In addition, some extra details regarding the nephroseq data has been added to the section “Assessment of gene expression profiles utilising additional datasets”

iv. The North Dublin Renal biobank data were adjusted for age and sex, but not apparently for the other clinical parameters corrected for in the epigenetic modelling

Authors reply: Yes, that is true. The kidney biopsy samples in Dublin were mainly derived from non-diabetic individuals with chronic kidney disease (CKD) with multiple etiologies and therefore adjustment e.g. for HbA1c was not considered relevant. Data on blood lipids (HDL cholesterol, triglycerides, LDL cholesterol) were incomplete preventing inclusion in the analyses. Smoking data were not collected. Thus, we decided to use only the standard covariates. We have also clarified the fact that these were CKD (not DKD) throughout the text.

Minor points

5. Can the authors comment on the choice of a combined analytic approach with an evaluation of “same direction of effect” in the two cohorts, rather than a discovery and validation cohort approach?

The authors reply: Instead of the discovery and validation approach, we wanted to use a larger cohort and complementary supporting data, where possible, to maximise power and reduce spurious findings. There was, however, a high agreement between the results of the two individual studies. Therefore, the discovery and validation approach would also have included most of the identified variants. We have now added the p-values (FDR adjusted) from the individual EWASs performed in the cohorts. All p-values in the minimal model EWASs were significant in both cohorts ($p < 0.05$), most being highly significant in both cohorts ($p < 10E-3$). The dmCpG with the most significant p-value was the same CpG in both cohorts (cg17944885). We also would like to emphasise that the genome-wide methylation levels in both cohorts were measured in the same laboratory, by the same personnel, using the same protocol and QC. Sample selection in terms of DNA quality and phenotyping was similar in both cohorts, which further impacted our decision to use the meta-analysis approach.

6. Can the authors present an analysis that demonstrates the additional discriminatory/predictive power of adding analysis of methylation status to existing clinical risk prediction algorithms (based on eGFR and proteinuria) in predicting eGFR decline?

Author’s reply: Indeed, it would be clinically relevant to know whether methylation at specific CpGs adds predictive power on top of other known risk factors for kidney failure. An earlier study in the FinnDiane cohort (Thorn et al. Diabetes Care 2015; doi: 10.2337/dc15-0641) listed several significant clinical risk factors for kidney failure: age, sex, age at diabetes onset, HbA1c, systolic blood pressure, serum triglycerides, retinal photocoagulation (as a mark of proliferative diabetic retinopathy), cardiovascular disease, and smoking. We have now tested the effect of adding eight of those clinical risk factors (age, sex, age at diabetes onset, HbA1c, systolic blood pressure, triglycerides, retinal photocoagulation, and smoking) as covariates to our survival model for kidney failure.

We examined the predictive power of the survival models with C-index (that is, concordance) using the R survival package `coxph()` function, and calculated the statistical difference between model C-indexes as suggested in <https://cran.r-project.org/web/packages/survival/vignettes/concordance.pdf>. This analysis was restricted to only those individuals with complete phenotypic data on all these variables ($n=387-388$). A model including only the clinical covariates had a C-index of 0.659. For 30/32 CpGs, the C-index improved when the CpG methylation was added to the model. For 13 CpGs, this improvement was significant ($p < 0.05$). The largest improvement was seen for the chr19 CpG cg17944885 (C-index: 0.745, compared to C-index of 0.659 from the model with clinical covariates only, $p = 5.6 \times 10^{-6}$ for model improvement).

Our original model included (baseline) age, sex and the 6 WCC as covariates. When we further added the baseline eGFR to the covariates, six CpGs stayed nominally associated ($p < 0.05$) with the progression to

kidney failure. Two CpGs on chr 19 locus retained their significance also at the Bonferroni corrected level for 32 CpGs (cg17944885 HR 95%CI = 1.40 [1.16, 1.70], $p=4.4 \times 10^{-4}$, $p_{\text{Bonf}}=0.014$; and cg25544931 HR 95%CI = 1.50 [1.18, 1.91], $p=0.0010$, $p_{\text{Bonf}}=0.033$).

This suggests that the results from the progression to kidney failure analysis largely reflect the association between baseline kidney function and CpG DNA methylation. Importantly, however, methylation at the locus on chr19 predicted the progression of DKD to kidney failure even on top of the baseline eGFR.

We have now added the following text into the “*Top-ranked dmCpGs and the development of kidney failure*” –paragraph:

“When we added the baseline eGFR to the survival model, the two CpGs on chr19 remained significantly associated with the risk of kidney failure (Bonferroni-adjusted $p<0.03$; Supplementary Table 4). Moreover, when adding CpG methylation to a survival model with eight clinical risk factors for kidney failure (age, sex, age at diabetes onset, systolic blood pressure, HbA1c, triglycerides, smoking, retinal photocoagulation) the model prediction improved significantly for 13/32 CpGs ($p<0.05$ for model concordance improvement).”

7. Did the authors consider censoring in the analysis of the 397 cases for whom follow up data was available? Associations with cardiovascular risk and hence death, and unavailability of follow up data may have been a source of bias.

Author’s reply: The 397 individuals with macroalbuminuria were followed up until kidney failure, death or December 2017. As all 397 individuals had a follow-up time, our data were right-censored.

Cox regression with right-censoring can be considered as one type of competing risk analysis, in contrast to the sub-hazard estimated e.g. with Fine and Gray competing risk analysis. In the presence of competing risk events, Cox regression may somewhat overestimate the results. While competing risk analysis would be particularly important for correct estimation of incidence, the Cox regression can be better suited for the identification of pathophysiological risk factors.

Nevertheless, we checked the mortality in our cohort: altogether 66 individuals were censored because of death. We performed the Fine and Gray competing risk analysis using kidney failure as an event of interest and death as a competing event and with the same covariates as in the Cox models. The results merely changed: 9/32 CpGs were associated with the risk of kidney failure (Bonferroni-corrected $p<0.05$) when in the Cox proportional-hazards model, the number of significant CpGs was 10. We believe this secondary analysis shows that Cox models are well suited for the prospective analysis in our study.

8. In the absence of new data, the subtitle for kidney tissue analysis (line 203) should make it clear that secondary analysis rather than new primary data are presented.

Author’s reply: We have now revised that title to: “Assessment of gene expression profiles in kidney tissue in additional external datasets”

Reviewer #2

Epigenome-wide meta-analysis identifies DNA methylation biomarkers associated with diabetic kidney disease

The study reports findings from an epigenomic-wide association study of type 1 diabetes kidney traits (DKD) using whole blood DNA methylation data from two cohorts: the United Kingdom-Republic of Ireland (UK-ROI) and Finland (FinnDiane) for a total of 1,304 participants (651 cases and 653 controls with and without DKD). Main findings are the association of DKD with 32 CpGs (31 dmCpGs from a minimally adjusted model and an additional CPG identified when adjusting for smoking). Of the 32 CpGs, 21 were also associated with kidney failure over time in the FinnDiane study. Replication was performed through a query of summary statistics from the CRIC study of individuals with type 2 diabetes. Several analyses to assess regulatory function were performed using datasets for transcription factors and gene expression in kidney tissue/cells, and a Mendelian randomisation identified evidence for causal effects for a CpG within the REV1 gene.

This is one of the largest EWAS of DKD in type 1 individuals of European ancestry and therefore it provides novel findings for this trait. However, the samples used for replication are likely not comparable to the discovery given CRIC participants have type 2 diabetes and the definition of DKD seems different. Given the difficulties of finding replication samples for type 1 diabetes, I suggest that the authors discuss the replication limitations in the discussion. This is important for future studies as these CpGs still need to be replicated.

Authors reply: We agree that finding suitable replication cohorts is challenging in type 1 diabetes as no other comparable large scale studies yet exist. Therefore, we opted for the largest possible discovery cohort and sought for supporting data for our findings from complementary studies. We have clarified in the text (methods) that the CRIC cohort was used to provide supporting data, not for independent replication:

“DKD-associated differentially methylated CpG-associations in external EWAS on DKD.

All 32 top-ranked dmCpGs were searched in available, non-overlapping summary EWAS data on DKD, which included an EWAS on albuminuria, eGFR, eGFR slope, and HbA_{1c} performed in 473 individuals with any diabetes from the CRIC study cohort”

Furthermore, we have added the following the limitations section:

There is an imperative need for larger cohorts with diabetes and epigenome-wide methylation data. Although some larger EWAS studies on kidney disease in the general population have been performed recently (Schlosser et al 2022), finding comparably sized cohorts with EWAS data and diabetes, particularly type 1 diabetes, remains challenging. In the lack of such studies, we were limited to finding additional support for our DKD-associated methylation differences in an external cohort with unspecified diabetes and a different phenotype for DKD (eGFR and albuminuria as quantitative measures), complemented by gene expression data.

The definition of DKD for the discovery sample needs to be clarified. The main text defines DKD cases as persistent macroalbuminuria, i.e. albumin 119 excretion rate (AER) > 300 mg/ml in urine, and controls had an AER within the normal range despite 120 a long duration of T1DM (≥15 years). However, the methods describe DKD as a low eGFR. Given replication was done for eGFR traits in CRIC, this needs to be clarified.

Authors reply: The definition of DKD was mainly based on albuminuria, i.e. macroalbuminuria was chosen as the threshold for DKD in both groups, as is standard for our GENIE cohort. A decline in eGFR usually

accompanies severely increased excretion of albumin into the urine. Thus, the participants in the macroalbuminuric group will have a low eGFR (under 60): even though we did not specifically require this in the FinnDiane cohort, 58.2% of the cases also have a eGFR lower than 60 (97% of controls in FinnDiane have a eGFR over 60). In the UK-Rol cohort, the primary definition of DKD is >10 years duration of T1DM with persistent macroalbuminuria, retinopathy, and hypertension; in this instance, cases also had eGFR<60 mL/min/1.73m². It is true that the CRIC study used a different phenotype (eGFR and albuminuria as continuous traits), and our aim was mainly to seek additional support from their data in lack of another cohort with type 1 diabetes and the exactly same phenotype (DKD). We have made the following changes to stress this in the discussion and the results:

In discussion: *“Although some larger EWAS studies on kidney disease in the general population have been performed recently, finding comparably sized cohorts with EWAS data and diabetes, particularly type 1 diabetes, remains challenging. In the lack of such studies, we were limited to finding additional support for our DKD-associated methylation differences in an external cohort with unspecified diabetes and a different phenotype for DKD (eGFR).”*

In results: *“Associations of top-ranked dmCpGs in external EWAS on DKD*

All 32 top-ranked dmCpGs were searched in available, non-overlapping summary EWAS data on DKD, which included in an EWAS on albuminuria, eGFR, eGFR slope, and HbA1c performed in 473 individuals with diabetes from the CRIC study cohort”

Related to the analyses, the case-control studies were matched on age, sex and other variables, so conditional analyses would be the more appropriate approach.

Authors reply: Yes, its correct that the case control studies were matched for age, sex and diabetes duration (and the FinnDiane study for smoking as well). We initially wanted to match for these variables, since factors such as age and smoking have a strong impact on methylation. Matching does however not completely control for the confounding factors, including those used to match cases with controls (Pearce et al 2015), so we decided to control for these factors in the adjusted models using a standard analysis method for epigenome-wide methylation data. The standard analysis (vs matched analysis) has slightly higher precision (Brokmeyer et al 1986) and is the best suitable method unless siblings or twin pairs have been chosen as case-controls (Pierce et al 2015). A comparison of age and sex matched and unmatched controls with long duration of type 1 diabetes and no evidence of kidney disease to cases with type 1 diabetes and end stage kidney disease, revealed minimal difference between the two models, with unmatched controls often enabling a larger sample size and therefore more power to identify risk loci (Smyth et al., 2021).

References

Pearce N. *Analysis of matched case-control studies* BMJ 2016; 352 :i969. <https://doi.org/10.1136/bmj.i969>

Brookmeyer R, Liang KY, Linet M. *Matched case-control designs and overmatched analyses.* Am J Epidemiol 1986;124:693-701. <https://doi.org/10.1093/oxfordjournals.aje.a114443>

Smyth, L J et al. “Assessment of differentially methylated loci in individuals with end-stage kidney disease attributed to diabetic kidney disease: an exploratory study.” Clinical epigenetics vol. 13,1 99. 1 May. 2021, doi:10.1186/s13148-021-01081-x

The longitudinal analysis methods are not described, included the covariates used in models.

Authors reply: We have now clarified the section's header that includes the analysis methods for the prospective analysis and the new header is : *“Top-ranked dmCpGs and longitudinal/survival analysis for DKD progression”* . We have also modified the text in that section to include the requested information:

“We extracted individual methylation M-values for the top dmCpGs (corrected $p \leq 9.9 \times 10^{-8}$) from the FinnDiane cohort case participants (n=401). The progression from macroalbuminuria to kidney failure (n=193 events) was studied using Cox proportional-hazards model with kidney failure as an outcome, methylation M-values as the predictor and with adjustment for age, sex and the six estimated WCCs. Bonferroni correction was applied for the association p-values.”

Clarify why principal components were not used in EWAS models.

Authors reply: To clarify, we have adjusted all the models for proportional white cell counts to account for some of the heterogeneity between and within the cohorts. We have not adjusted the models for genetic principal components, as one would typically do in a GWAS study, since genetically derived population structure differences have a negligible effect on genome-wide methylation levels. We acknowledge that some previous large multi-ethnic EWAS studies in the general population have adjusted for the genetic PCs, but the EWAS published, for example, in DKD, have not. Including PCs i.e. up to 10 additional covariates may also reduce study power. For these reasons, we decided to instead focus on a stringent and harmonised quality control of the raw data and other adjustments, to maximise the reliability of our results. Adjustment for technical PCs such as bisulfite batch, plate, sentrix position and control probes was implemented . Furthermore, there was a considerable overlap between association results from the two studies; for example, the same top hit was observed in the two individual EWAS studies and all the identified variants had a p-value <0.05 (FDR adjusted) in the individual EWAS:s.

It seems that the author used Methylation M-values for analysis, but it is not clear if methylation was used as a predictor or as an outcome.

Authors reply: Yes, the methylation M values are the predictors, and the outcome is the kidney failure event. We apologise for not being clear on this point, and we have now added this information to the methods. Please see methods:

“The progression from macroalbuminuria to kidney failure (n=196 events) was studied using Cox proportional-hazards model with kidney failure as an outcome, methylation M-values as the predictor and with adjustment for age, sex, and the six estimated WCCs.”

Methods used for differential methylation levels within regions and gene-based analysis described in results in the paragraph starting at line 257 are not described in detail in methods.

Authors reply: We have now described the gene-based analysis in more detail in the methods:

“In RnBeads, the region-based analyses are calculated as the average difference in means across all CpGs in a region of the two groups being compared (here DKD vs controls) and a combined p-value is calculated from all CpGs p-values in the region (Makambi et al., 2003).”

References:

Makambi, K. (2003) Weighted inverse chi-square method for correlated significance tests. Journal of Applied Statistics, 30(2)

The authors need to report lambdas for EWAS.

Authors reply: We have previously reported the genomic inflation factor for our cohorts as part of a GWAS, with lambdas very close to 1.00 (Salem et al., 2019), so population substructure is not a major issue in our study. The epigenome-wide methylation data, however, is very different from the genome-wide genetic data as methylation levels are quantitative data that show stronger associations with phenotypic traits or diseases than a genotypic marker, which do not change in response to the environment (van Iterson et al., 2017). The methylation levels are instead more susceptible to confounding by technical batches and biological influences, such as cell type heterogeneity, which we adjusted for in our analyses. In addition, we applied a dual adjustment for multiple testing, which included both FDR adjustment of the p-values prior meta-analysis and a setting threshold for epigenome-wide significance in the meta-analysis. Consequently, the lambdas in the EWAS here are deflated : 0.60 (minimal model), 0.40 (minimal model adjusted for smoking) and 0.11 (Maximal model). With this background and with the knowledge that the genomic inflation factor lambda is not suitable for measuring inflation in EWAS (van Iterson et al., 2017), the lambda values are not included in the text.

References:

van Iterson, M., van Zwet, E.W., the BIOS Consortium. et al. Controlling bias and inflation in epigenome- and transcriptome-wide association studies using the empirical null distribution. *Genome Biol* 18, 19 (2017). <https://doi.org/10.1186/s13059-016-1131-9>

Figure 1 C, qq plot distribution needs some comment.

Authors reply: Yes, we have added the following comment to the results section (Meta-analysis of dmCpGs from RnBeads analysis):

“The QQ plots showed no presence of inflation (Figure 1), but instead we observed some deflation of the p-values, particularly in the maximal model (Figure 1c), which was likely due to the FDR-adjustment of the p-values and the multiple adjustments included in that model.”

Mendelian randomisation results do not include analysis to verify assumptions.

Authors reply: Thank you for raising this point, we have now added analyses to verify the assumptions in the MR analysis (test of the Egger intercept and heterogeneity p-value, supplementary Table 11). These analyses require that more than two SNPs (mQTLs) are available for the exposure (methylation). Unfortunately, only three of the dmCpGs had more than two independent mQTLs (n SNPs ranged from 1 to 4 for the CpGs with mQTLs). For these three, we found no evidence of pleiotropic effects being present. In addition, to assure that the SNPs were robustly associated with the exposure, we selected only instruments that were highly ($p < 10^{-5}$ used as the threshold, but all SNPs were highly associated with the methylation level at each respective site, $p < 10^{-30}$) and independently ($r^2 > 0.8$) associated with the exposure (methylation levels at the specific site). Both the text in the methods and the text in the results have been modified to address this question:

In methods: “We searched mQTL variants for each CpG from results from the Genetics of DNA Methylation Consortium (GoDMC)²¹, and selected both cis and trans meQTLs available on the GoDMC site, but only those that were independently associated ($p < 10^{-5}$, $r^2 > 8.0$) with methylation at each CpG site (mqtlb.godmc.org.uk). We retrieved the genetic variant-outcome associations from the latest genome-wide association study on DKD in T1DM⁶⁷. Causal ORs were calculated for each CpG using the Wald ratio test (one mQTL) or the inverse variance weighted method (mQTL > 1) implemented in the R package TwoSampleMR. If several mQTLs (n > 2) were identified for the methylation levels at a CpG, we performed

additional tests (heterogeneity and the Egger intercept test) to address potential pleiotropy, which could violate the instrumental variable assumptions.”

In results: “For CpGs with several instruments available (mQTLs for CpGs in or near ADPRHL1, INPP4B and ZNF625-ZNF20), we found no evidence of pleiotropy using the heterogeneity or Egger intercept test (Supplementary Table 11).”

Table 1, add eGFR values for the studies. Table 2, add footnote that the CRIC study has participants with T2D CKD

Authors reply: The information has now been added to the tables. The CRIC study does not specify the type of diabetes (could include both types of diabetes), so we added the DKD to CRIC and DKD-T1DM to our prospective analysis column.

What are the differences in results reported in Figure 1 and Figure 3?

Authors reply: Figure 1 shows the association results (p-values) versus the chromosomal position. The Volcano plot (Figure 3), shows the association results (also p-values) versus the mean methylation difference (i.e., the beta methylation levels in cases minus the beta methylation level in controls) in cases vs controls. We have now modified the legends to clarify this further:

“Figure 3 Volcano plot of methylation differences for CpGs (DKD cases vs controls without DKD) in the meta-analyses of the minimal model (a), minimal model + smoking (b), and maximal model (c). The red colour denotes dmCpGs with $p < 9.9 \times 10^{-8}$ and hypermethylated in DKD cases vs controls and the blue colour denotes dmCpGs with $p < 9.9 \times 10^{-8}$ and hypomethylated in DKD cases vs controls. The x-axis shows the mean methylation difference (the difference in methylation β -values) and the y-axis shows $-\log_{10}(p\text{-values})$. The dotted line indicates the epigenome-wide significance threshold ($p \leq 9.9 \times 10^{-8}$).”

The data sharing section is missing.

Authors reply: This has now been added to the text. We will share the summary data but the local legislations and the consents do not allow sharing of individual-level data.

“Data availability

The epigenome-wide DNA methylation summary data from the meta-analysis (all three models) are available at <https://www.qub.ac.uk/sites/GenPECT/>. All other relevant data related to this study are available within the article or in its supplement (link to supplement). Individual-level data is not available due to local legislation.”

The authors could state how their results move the field forward and their view of future studies.

Authors reply: Thank you for this suggestion, we modified the conclusion section according to the suggestion:

“In summary, this EWAS meta-analysis performed on two well-characterized cohorts identified several epigenetic signatures associated with DKD in T1DM and highlighted several new genes, not previously associated with kidney disease in T1DM. These findings provide future studies with some potential links between the genes and the milieu in DKD. The prospective analyses additionally showed that these epigenetic signatures have the potential to identify individuals at increased risk of DKD and progressing to kidney failure. As they are potentially reversible, they offer the possibility of therapeutic intervention, if they prove to be

causal for DKD, as the dmCpG located in the REV1 gene did. More comprehensive meQTL data is needed to allow more systematic MR analyses for causality as well as functional studies assessing the biological underpinnings of these loci”

REVIEWERS' COMMENTS

Reviewer #1 (Remarks to the Author):

The authors have addressed all of my points comprehensively

Reviewer #2 (Remarks to the Author):

Appropriate answers to the reviewer's comments. Two additional important published studies of DNA methylation and kidney traits should also be acknowledged for completeness (PMID: 35710981, PMID: 33931109).

RESPONSE TO REVIEWERS' COMMENTS

Reviewer #1 (Remarks to the Author):

The authors have addressed all of my points comprehensively

Response to Reviewer #1: Thank you

Reviewer #2 (Remarks to the Author):

Appropriate answers to the reviewer's comments. Two additional important published studies of DNA methylation and kidney traits should also be acknowledged for completeness (PMID: 35710981, PMID: 33931109).

Response to Reviewer #2: Thank you, we agree that these are two important papers in the field of DNA methylation and kidney traits. The Liu et al paper (PMID: 35710981) should have been included, as we performed lookups in their data, but by mistake it was not included. This mistake has now been corrected. The breeze paper (PMID: 33931109) has also been included now in the discussion (ref nr 27): *“Hypermethylation at cg17944885 has previously been associated with reduced eGFR in males with the human immunodeficiency virus²⁴, as well as reduced eGFR and increased risk of CKD and albuminuria in an EWAS meta-analysis in the general population^{25–27”}.*